# Specific targeting of inflammatory osteoclastogenesis by the probiotic yeast *S. boulardii* CNCM I-745 reduces bone loss in osteoporosis

Maria-Bernadette Madel[1,2‡], Julia Halper[1,2‡], Lidia Ibáñez[3], Lozano Claire[4], Matthieu Rouleau[1,2], Antoine Boutin[1,2], Adrien Mahler[1,2], Rodolphe Pontier-Bres[2,5], Thomas Ciucci[6§], Majlinda Topi[1,2], Christophe Hue[7], Jerome Amiaud[8], Salvador Iborra[9], David Sancho[10], Dominique Heymann[11], Henri-Jean Garchon[7,12], Dorota Czerucka[2,5], Florence Apparailly[4], Isabelle Duroux-Richard[4], Abdelilah Wakkach[1,2†], Claudine Blin-Wakkach[1,2*†]

[1]Université Côte d'Azur, CNRS, LP2M, Nice, France; [2]LIA ROPSE, Laboratoire International Associé Université Côte d'Azur - Centre Scientifique de Monaco, Nice and Monaco, France; [3]Department of Pharmacy, Cardenal Herrera-CEU University, Valencia, Spain; [4]IRMB, Université Montpellier, Montpellier, France; [5]Centre Scientifique, de Monaco, Monaco; [6]Laboratory of Immune Cell Biology, Center for Cancer Research, National Cancer Institute, National Institutes of Health, Bethesda, United States; [7]Université Paris-Saclay, UVSQ, INSERM, Infection et inflammation, Montigny-Le-Bretonneux, France; [8]Inserm, Universite de Nantes, Nantes, France; [9]Department of Immunology, Ophthalmology and ENT. School of Medicine, Universidad Complutense de Madrid, Madrid, Spain; [10]Immunobiology Laboratory, Centro Nacional de Investigaciones Cardiovasculares (CNIC), Madrid, Spain; [11]Université de Nantes, Institut de Cancérologie de l'Ouest, Saint Herblain, France; [12]Genetics Division, Ambroise Paré Hospital, AP-HP, Boulogne-Billancourt, France

**\*For correspondence:**
claudine.blin@univ-cotedazur.fr (CB-W);
claudine.blin@univ-cotedazur.fr (CB-W)

†These authors contributed equally to this work
‡These authors also contributed equally to this work

**Present address:** §David H. Smith Center for Vaccine Biology and Immunology, Department of Microbiology and Immunology, University of Rochester Medical Center, Rochester, United States

**Abstract** Bone destruction is a hallmark of chronic inflammation, and bone-resorbing osteoclasts arising under such a condition differ from steady-state ones. However, osteoclast diversity remains poorly explored. Here, we combined transcriptomic profiling, differentiation assays and in vivo analysis in mouse to decipher specific traits for inflammatory and steady-state osteoclasts. We identified and validated the pattern-recognition receptors (PRR) Tlr2, Dectin-1, and Mincle, all involved in yeast recognition as major regulators of inflammatory osteoclasts. We showed that administration of the yeast probiotic *Saccharomyces boulardii* CNCM I-745 (*Sb*) in vivo reduced bone loss in ovariectomized but not sham mice by reducing inflammatory osteoclastogenesis. This beneficial impact of *Sb* is mediated by the regulation of the inflammatory environment required for the generation of inflammatory osteoclasts. We also showed that *Sb* derivatives as well as agonists of Tlr2, Dectin-1, and Mincle specifically inhibited directly the differentiation of inflammatory but not steady-state osteoclasts in vitro. These findings demonstrate a preferential use of the PRR-associated costimulatory differentiation pathway by inflammatory osteoclasts, thus enabling their specific inhibition, which opens new therapeutic perspectives for inflammatory bone loss.

## Editor's evaluation

This important work substantially advances our understanding osteoclast diversity such as inflammatory and state osteoclasts in pathological condition, which was demonstrated by administration of the yeast probiotic Saccharomyces boulardi CNCM I-745 (Sb) in vivo reduced bone loss in OVX but not sham mice by reducing inflammatory osteoclasts, and Sb derivatives specifically inhibited directly the differentiation of inflammatory but not steady state osteoclasts in vitro. The evidence supports the conclusions, through combined transcriptomic profiling, differentiation assays and in vivo analysis in mice to decipher specific traits for inflammatory and steady state osteoclasts. The work will be of broad interest to Immunologists, Microbiologists and Cell biologists.

## Introduction

Osteoclasts (OCLs) are multinucleated phagocytes derived from monocytic progenitors and specialized in bone resorption (*Madel et al., 2019*). Similar to other cells of monocytic origin, they are also innate immunocompetent cells and heterogeneous in their phenotype, function, and origin (*Ibáñez et al., 2016*; *Jacome-Galarza et al., 2019*; *Madel et al., 2020*; *Madel et al., 2019*; *Yahara et al., 2020*). OCLs derived from steady-state bone marrow (BM) cells or from BM CD11b$^+$ monocytic cells (MN-OCLs) promote tolerance by inducing CD4$^+$ and CD8$^+$ regulatory T cells (tolerogenic OCLs [t-OCLs]) (*Ibáñez et al., 2016*; *Kiesel et al., 2009*). In contrast, in the context of bone destruction linked to inflammatory bowel disease (IBD) or when derived from dendritic cells (DC-OCLs), OCLs induce Tnf-α-producing CD4$^+$ T cells (inflammatory OCLs [i-OCLs]) (*Ibáñez et al., 2016*; *Madel et al., 2020*). OCLs associated with inflammation can be identified by expression of Cx3cr1 (fractalkine receptor) and the proportion of Cx3cr1$^+$ OCLs increases in osteoporosis, IBD, and after Rank-L treatment (*Ibáñez et al., 2016*; *Madel et al., 2020*). However, Cx3cr1 is only expressed in approximately 20% of i-OCLs (*Ibáñez et al., 2016*; *Madel et al., 2020*), which highlights their heterogeneity while limiting the possibility to analyze them in the context of pathological bone loss and urges the identification of novel markers.

Current anti-resorptive therapies aim to globally inhibit OCLs, without considering their recently established diversity. In the long term, they result in poor bone remodeling which may increase the risk of atypical fractures (*Reyes et al., 2016*). Therefore, an in-depth characterization of OCLs associated with healthy versus inflammatory bone resorption would allow the identification of distinct characteristics that could help to specifically target i-OCL.

i-OCLs arise under the control of persistent high levels of Rank-L, IL-17, and TNFα mainly produced by CD4$^+$ T cells that play a major role in pathological osteoclastogenesis observed in osteoporosis and IBD (*Cenci et al., 2000*; *Ciucci et al., 2015*; *Ibáñez et al., 2016*; *Li et al., 2011*). Interestingly, the emergence of such osteoclastogenic CD4$^+$ T cells is associated with gut dysbiosis and increased intestinal permeability (*Jones et al., 2018*; *Li et al., 2016*). In line with this, bacterial probiotics such as *Lactobacillus* and *Bifidobacteria* have been shown to effectively reduce osteoporotic bone loss (*Li et al., 2016*; *Britton et al., 2014*; *Ohlsson et al., 2014*; *Ibáñez et al., 2019b*), but their specific effect on i-OCLs remains unknown.

Here, using a comparative RNAseq approach performed on sorted pure mature MN-OCLs and DC-OCLs as models of t-OCLs and i-OCLs, respectively, as already established (*Ibáñez et al., 2016*; *Madel et al., 2020*), we showed that the two OCL populations are distinctly equipped to respond to different signals that can modulate their differentiation. In particular, the pattern recognition receptors (PRRs) Dectin-1, Tlr2, and Mincle involved in the response to fungi (*Li et al., 2019*; *Sancho and Reis e Sousa, 2012*) are overexpressed in i-OCLs. Administration of the probiotic yeast *Saccharomyces boulardii* CNCM I-745 (*Sb*), which is used in the treatment of gastrointestinal disorders for its anti-inflammatory properties and its capacity to restore the gut microbiota (*Czerucka and Rampal, 2019*; *Terciolo et al., 2019*), significantly reduces bone loss and inflammatory parameters in vivo in ovariectomized (OVX) mice. In vitro, Sb derivates as well as low doses of agonists of the PRRs overexpressed in i-OCLs specifically inhibit the differentiation of these cells without affecting t-OCLs. These data open perspectives on targeting specific OCL populations and provide evidence for the protective effect of a probiotic yeast on inflammatory bone resorption. Our study unveils very new insights into the regulation and modulation of i-OCLs and enables a better understanding of the molecular mechanisms involved in inflammation-induced bone erosions.

## Results

### Transcriptomic profiling reveals upregulation of innate immune receptors in i-OCLs

To better understand the differences between t-OCLs and i-OCLs, we performed a comparative RNA-sequencing (RNAseq) analysis between sorted mature (≥three nuclei) MN-OCLs (originating from BM CD11b$^+$ monocytic cells) and DC-OCLs (differentiated from BM-derived DCs) representing t-OCLs and i-OCLs, respectively, as already demonstrated (*Ibáñez et al., 2016*; *Madel et al., 2020*; *Figure 1—figure supplement 1a–b*). A total of 906 genes (log$_2$FC (Fold Change) ≥1; p<0.05) were significantly differentially expressed between the two OCL subsets (*Figure 1a*), including *Cx3cr1* previously identified as a marker of i-OCLs (*Ibáñez et al., 2016*; *Madel et al., 2020*). The most differentially expressed genes were related to innate immunity and immune defense responses (*Figure 1b–d*), confirming our previous observation that i-OCLs and t-OCLs differ in their immune capacity (*Ibáñez et al., 2016*).

Volcano plot representation showed that the expression of genes involved in OCL resorbing activity as well as in the Rank-Rank-L differentiation pathway was not significantly different in the two OCL populations, except for *Nfatc1*. However, major differences were found in genes associated with the Ig-like receptor-dependent costimulatory OCL differentiation pathway (*Koga et al., 2004*; *Merck et al., 2004*; *Figure 1e*). Interestingly, annotation of these genes (EnrichR/Kegg analysis) highlighted that they are also linked to pathways related to immune responses and PRRs, notably the C-type lectin receptors (CLRs) (*Figure 1f*).

These results were confirmed by a global miRNA profiling on the same sorted mature OCL samples (≥three nuclei). Volcano plot visualization confirmed that the two OCL subsets differ also in their miRNA expression pattern (*Figure 1g*). Differentially expressed miRNAs (FC ≥2, p<0.05, quantitative PCR [qPCR] cycle threshold [CT] values <32) (*Table 1*) were further validated by quantitative real-time PCR (RT-qPCR) confirming that *Mir151*, *Mir342*, *Mir146b,* and *Mir155* were significantly upregulated in i-OCLs, and that *Mir185*, *Mir674*, *Mir26b,* and *Mir29a* were upregulated in t-OCLs (*Figure 1h*). Using the miRWalk software that provides information on miRNA-target interactions (*Dweep and Gretz, 2015*), we carried out an integrative analysis of these miRNAs and genes that were significantly differentially expressed between both OCL subsets to find possible relationships (*Table 2*). In agreement with the RNAseq analysis, computational analysis of these related genes (EnrichR/ Kegg analysis) also revealed an association with the OCL differentiation pathway as well as the C-lectin-like receptor pathway (*Figure 1i*).

These data strongly suggest that CLRs could play an important role in the specific properties of i-OCLs and t-OCLs. Analysis of the RNAseq data for the expression of genes involved in the CLR and TLR signaling pathways confirmed major differences between t-OCLs and i-OCLs (*Figure 1j*). Validation by flow cytometry analysis of mature multinucleated OCLs confirmed that the proportion of OCLs expressing Dectin-1, Mincle, and Tlr2, but not Dectin-2, was significantly higher in i-OCLs compared to t-OCLs (*Figure 1k*). We also analyzed their expression in OCL progenitors. The proportion of cells expressing Dectin-1, Mincle, and Tlr2 was much higher in CD11c$^+$ BM-derived DCs than in BM CD11b$^+$ MNs, and the proportion of i-OCL progenitors expressing Dectin-2 was lower than for the other markers (*Figure 1l*).

### The probiotic yeast *Saccharomyces boulardii* CNCM I-745 protects from osteoporosis-induced bone loss in vivo

The herein investigated OCL populations were derived from purified progenitor cells which are a powerful approach to identify major differences (*Ibáñez et al., 2016*; *Madel et al., 2020*), while in vivo, OCLs arise from a mixture of different BM progenitors whose proportions are depending on the pathophysiological conditions (*Madel et al., 2019*). Thus, we compared BM-derived OCLs from OVX-induced osteoporotic mice in which the proportion of i-OCL increases (*Madel et al., 2020*) to OCLs derived from SHAM mice. FACS analysis revealed higher proportions of Dectin-1$^+$ and Tlr2$^+$ OCLs, in OCLs generated from OVX mice compared to those from SHAM mice, while the proportion of Mincle$^+$ OCLs was not significantly altered (*Figure 2a*).

These PRRs share the capacity to sense fungi and control responses to pathogenic or commensal gut mycobiome (*Li et al., 2019*; *Sancho and Reis e Sousa, 2012*). Therefore, we treated SHAM and OVX mice by oral gavage with the probiotic yeast *Sb* (*Figure 2b*). As expected, OVX mice showed

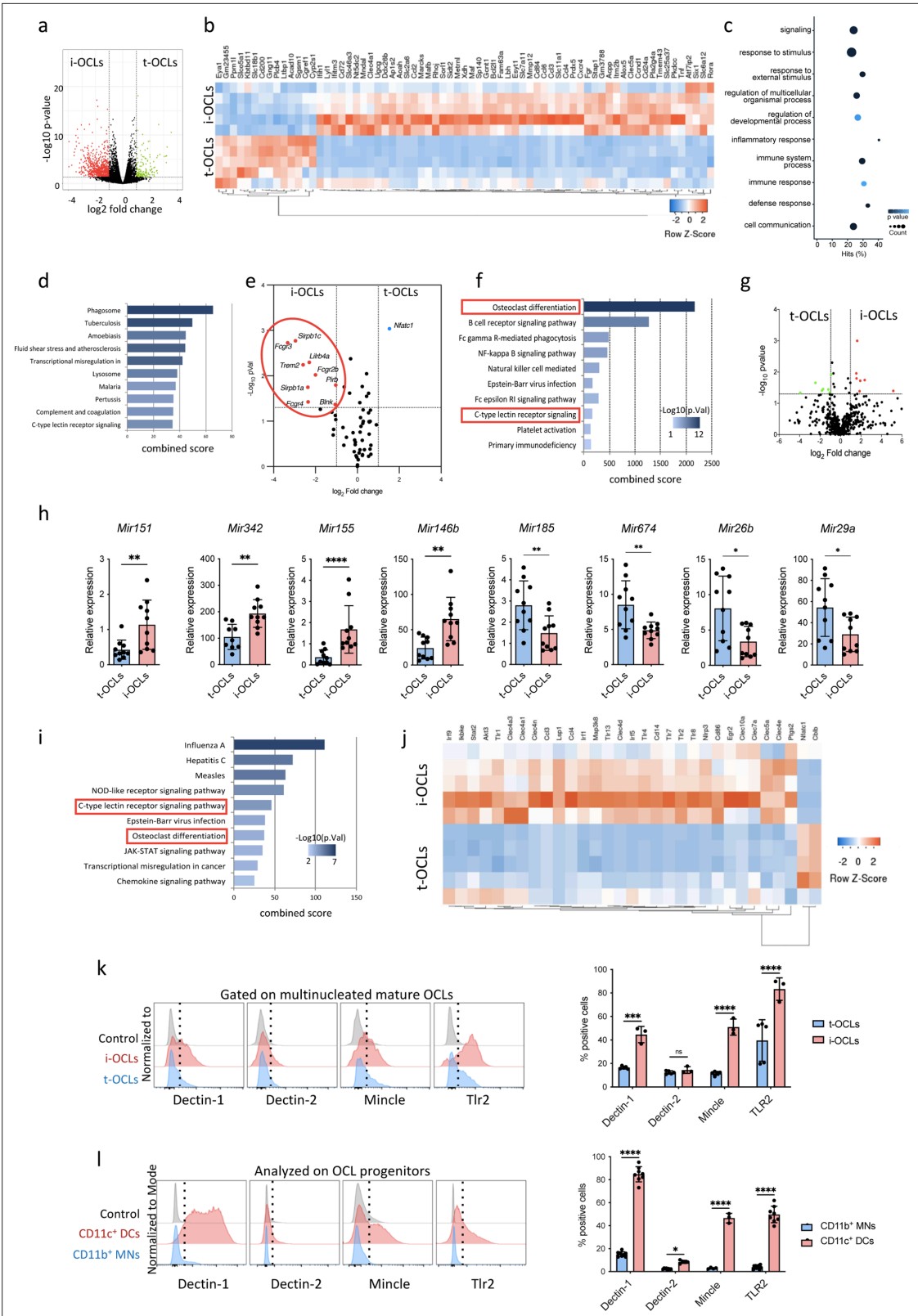

**Figure 1.** Comparative transcriptomic analysis of inflammatory osteoclasts (i-OCLs) versus tolerogenic OCLs (t-OCLs) reveals two distinct populations of OCLs and differences in their differentiation pathway. (**a**) Volcano plot analysis of differentially expressed (DE) genes (p<0.05; FC ≥2) between sorted mature i-OCLs (dendritic cells [DC]-derived OCLs) and t-OCLs (monocytic cells [MN]-derived OCLs) differentiated and sorted (≥ three nuclei) as presented in **Figure 1—figure supplement 1**. Analysis was performed on n=5 biological replicates for each group. (**b**) Heatmap visualization of the

*Figure 1 continued on next page*

*Figure 1 continued*

z-scored expression of the top 70 genes (p<0.05; FC ≥2) that are significantly DE between the two OCL subsets. (**c**) Gene ontology analysis of biological processes associated with DE genes. (**d**) EnrichR annotation (Kegg) for the 906 DE genes between MN-OCLs and DC-OCLs. (**e**) Transcriptomic analysis of selected genes (from Kegg mmu04380, osteoclast differentiation) involved in bone resorption (*Acp5, Car2, Clcn7, Ctsk, Mmp9, Ostm1, Tcirg1*), the Rank differentiation pathway (*Chuk, Ikbkb, Ikbkg, Map2k1, Map2k7, Map3k7, Mapk8, Mapk9, Nfatc1, Nfkb1, Nfkb2, Tab1, Tab2, Tnfrsf11a, Traf2, Traf6*), and the co-stimulatory differentiation pathway (*Blnk, Fcgr1, Fcgr2b, Fcgr3, Fcgr4, Lilrb4a, Oscar, Plcg2, Sirpb1b, Sirpb1c, Syk, Trem2, Tyrobp*) in i-OCLs and t-OCLs on n=5 biological replicates per group. (**f**) EnrichR annotation (Kegg) for the genes involved in the OCL co-stimulatory differentiation pathway (from **d**). (**g**) Volcano plot visualization of the comparative miRNome analysis of i-OCLs and t-OCLs performed on n=5 biological replicates for each group. (**h**) Quantitative real-time PCR analysis (mean ± standard deviation (SD)) on t-OCLs and i-OCLs (n=10 biological replicates per group). miRNA expression was normalized to the *sno202* expression using the $2^{-(\Delta Ct)}$ method. (**i**) EnrichR annotation (Kegg) for the target genes (from **Table 2**) of the DE miRNAs. (**j**) Heatmap visualization of the RNAseq data for selected genes involved in the C-type lectin receptor and TLR (Toll Like Receptor) signaling pathway (from Kegg) in i-OCLs and t-OCLs. (**k**) FACS (Flow Cytometry Cell Sorting) histograms and quantification (mean ± SD) of mature i-OCLs and t-OCLs for the expression of Dectin-1 and 2, Tlr2, and Mincle. Mature OCLs were gated as shown in **Figure 1—figure supplement 1b**. (**l**) FACS analysis on BM-derived CD11c⁺ DCs, and CD11b⁺ BM cells (used as OCL progenitors) and quantification (mean ± SD) of positive cells for the expression of Dectin-1 and 2, Tlr2, and Mincle each marker. *p<0.05; **p<0.01; ***p<0.001; ****p<0.0001.

The online version of this article includes the following figure supplement(s) for figure 1:

**Figure supplement 1.** Cell preparation and gating strategies.

atrophy of the uterus and their body weight increased compared to SHAM control mice, which was not affected by treatment with *Sb* (**Figure 2—figure supplement 1a–b**). Micro-computed tomographic (µCT) analysis revealed that *Sb*-treated OVX mice displayed reduced trabecular bone loss, as assessed by a significantly higher trabecular bone volume fraction (BV/TV) and trabecular number and less significant reduction in trabecular separation compared to untreated OVX mice (**Figure 2c–d**). Furthermore, administration of *Sb* resulted in a reduction of TRAcP⁺ OCL surface per bone surface in OVX mice (**Figure 2e–f**) and a significantly reduced serum level of cross-linked C-telopeptides of type I collagen (CTX), confirming a decrease in OCL activity (**Figure 2g**). Importantly, *Sb* treatment did not affect cortical parameters (**Figure 2—figure supplement 1b**), which could be due to a too short treatment period, as suggested in previous studies using bacterial probiotics (**McCabe et al., 2013**;

**Table 1.** List of the miRNAs differentially expressed between t-OCLs and i-OCLs.

| Official symbol | Official name | CT value | Log 2 (FC) | p-Value |
|---|---|---|---|---|
| *Mir151* | *mmu-miR-151–3 p* | 28.3 | –5.06 | 0.04 |
| *Mir342* | *mmu-miR-342–3 p* | 23.1 | –2.40 | 0.018 |
| *Mir351* | *rno-miR-351* | 29.1 | –1.94 | 0.019 |
| *Mir125a* | *mmu-miR-125a-3p* | 30.2 | –1.84 | 0.041 |
| *Mir146b* | *mmu-miR-146b* | 21.3 | –1.64 | 0.001 |
| *Mir155* | *mmu-miR-155* | 26.5 | –1.56 | 0.011 |
| *Mir30e* | *mmu-miR-30e* | 21.4 | 0.59 | 0.03 |
| *Mir106a* | *mmu-miR-106a* | 17.8 | 0.70 | 0.011 |
| *Mir26b* | *mmu-miR-26b* | 20.9 | 0.70 | 0.024 |
| *Mir17* | *mmu-miR-17* | 17.0 | 0.82 | 0.005 |
| *Snord65* | *snoRNA135* | 18.4 | 0.90 | 0.029 |
| *Mir29a* | *mmu-miR-29a* | 19.1 | 0.93 | 0.012 |
| *Mir674* | *mmu-miR-674* | 20.6 | 1.06 | 0.045 |
| *Mir185* | *mmu-miR-185* | 23.8 | 1.20 | 0.036 |
| *Mir18a* | *mmu-miR-18a* | 22.7 | 1.71 | 0.035 |
| *Mir148a* | *mmu-miR-148a* | 23.7 | 1.82 | 0.039 |
| *Mir32* | *mmu-miR-32* | 28.6 | 2.45 | 0.022 |
| *Mir130a* | *mmu-miR-130a* | 25.2 | 3.94 | 0.046 |

**Table 2.** Putative targeting of the differentially expressed genes by the eight validated miRNAs (in *Figure 1h*).
This table lists the 174 differentially expressed genes putatively targeted by one or more discriminant miRNAs according to the in silico prediction analysis using four algorithms (TargetScan, Miranda, RNA22, and miRWalk). Crosses indicate a putative target predicted by three or four algorithms (score 3–4). i-OCLs: inflammatory osteoclasts; t-OCLs: tolerogenic osteoclasts.

| Gene symbol | -log(p-value) | Fold change (log2) | i-OCLs miRNAs (score 3–4) | | | | t-OCLs miRNAs (score 3–4) | | | |
|---|---|---|---|---|---|---|---|---|---|---|
| | | | Mir155 | Mir146b | Mir342 | Mir151 | Mir185 | Mir674 | Mir26b | Mir29a |
| 1810011H11Rik | 2,78 | –3,4 | | | | X | | | X | X |
| 5430427O19Rik | 2,56 | –4,0 | | X | | | | | | X |
| 9930111J21Rik1 | 3,31 | –3,5 | X | | | | | | | |
| A530064D06Rik | 3,67 | –4,2 | | | | X | | | | |
| Abcc3 | 2,41 | –3,0 | | | | | | | | X |
| Acpp | 2,81 | –2,4 | | | | X | | X | | X |
| Acvrl1 | 2,48 | –2,8 | | | | X | X | X | | |
| Adap2 | 3,93 | –3,2 | | | | | X | | | |
| Ahrr | 2,37 | –2,7 | | | | | X | X | | X |
| Akt3 | 3,24 | –2,5 | | X | X | X | X | X | | X |
| Alox5 | 2,86 | –2,5 | | | | | | X | | |
| Als2cl | 3,45 | –4,5 | X | X | | X | X | X | | |
| Aoah | 4,41 | –4,6 | | | | | X | | | |
| Apobec1 | 3,21 | –2,9 | X | | | | X | | | |
| Arhgap15 | 3,01 | –3,7 | X | | X | | | X | | |
| Arhgap19 | 2,35 | –2,6 | | | X | | | X | | |
| Arhgef10l | 2,80 | –2,8 | | | X | | | | | |
| Arl4c | 3,00 | –2,6 | | | X | | | | X | |
| Atf3 | 2,61 | –3,6 | | | | | | X | | |
| Atp1a3 | 3,36 | –2,6 | | | | | X | X | | |
| B430306N03Rik | 3,14 | –3,4 | | X | X | | X | X | | X |
| B4galt6 | 2,90 | –3,4 | X | X | X | | X | X | | X |
| Bhlhe41 | 3,45 | –3,8 | | X | | X | X | X | X | X |
| Bst1 | 2,49 | –3,5 | | X | | | | | | |
| Btg2 | 2,67 | –3,5 | | X | | | X | X | | X |
| C3ar1 | 2,90 | –3,0 | | X | | | X | X | | |
| C5ar1 | 3,30 | –3,9 | | | | | | | X | X |
| Camk1d | 2,47 | –3,1 | X | X | | X | X | X | X | X |
| Ccl6 | 5,25 | –5,6 | | | | | X | | | |
| Cd28 | 2,56 | –1,9 | | X | | X | X | | | |
| Cd300lb | 3,16 | –3,6 | | X | | | | | | X |
| Cd300ld | 4,11 | –4,8 | | | | | X | X | | |
| Cd36 | 4,26 | –4,0 | | | X | | X | | | |
| Cd5l | 3,43 | 3,4 | | | | | X | | | |
| Cd93 | 3,69 | –4,0 | X | X | X | | X | X | X | X |
| Cdc42ep3 | 3,01 | –3,1 | | | | | X | | | |

*Table 2 continued on next page*

*Table 2 continued*

| Gene symbol | -log(p-value) | Fold change (log2) | i-OCLs miRNAs (score 3–4) | | | | t-OCLs miRNAs (score 3–4) | | | |
| --- | --- | --- | --- | --- | --- | --- | --- | --- | --- | --- |
| | | | Mir155 | Mir146b | Mir342 | Mir151 | Mir185 | Mir674 | Mir26b | Mir29a |
| Cebpa | 2,99 | –2,7 | | | | | | | X | |
| Chst12 | 2,72 | –2,6 | | | | | X | | | |
| Clec4a1 | 3,89 | –2,7 | | | | | X | | | |
| Clec4n | 3,02 | –3,8 | | | | | | | X | |
| Cmpk2 | 3,05 | –2,8 | | | | X | | | | |
| Cnr2 | 2,91 | –3,9 | | | | | X | X | | |
| Cybb | 4,24 | –4,0 | | | | | | | X | |
| Dab2 | 3,30 | –4,0 | X | | | | X | | X | X |
| Dapk1 | 2,44 | –1,8 | | | X | | | | X | |
| Ddx58 | 3,39 | –3,1 | | | | | | | | X |
| Dhrs3 | 3,35 | –3,2 | | | | X | | | | |
| Dock1 | 2,35 | –1,8 | | | X | | X | | | |
| Dock2 | 2,65 | –2,7 | | | | | X | | | |
| Dram1 | 2,98 | –2,6 | X | | X | | X | X | X | X |
| Dtx3l | 3,00 | –2,4 | X | | | | X | | | |
| Ednrb | 3,79 | –3,7 | | X | X | | | | | X |
| Elmsan1 | 2,86 | –2,4 | | X | | | | X | | X |
| Entpd1 | 2,84 | –2,5 | | X | | X | | | | |
| Eps8 | 3,85 | –4,9 | | | | | | | X | |
| Fabp5 | 3,23 | –2,3 | | X | | | | | | |
| Fads1 | 4,20 | –3,3 | | | | | X | | | |
| Fcgr4 | 3,58 | –3,2 | | | | | X | | | |
| Frmd4b | 2,92 | –2,2 | | | | | | | X | |
| Gas7 | 2,43 | –2,5 | | X | | | X | X | | X |
| Gbp2 | 3,70 | –3,3 | | | | | X | | | |
| Gcnt1 | 2,37 | –2,8 | | | | | X | | | |
| Ggta1 | 3,34 | –3,1 | X | | X | | | | | |
| Gna15 | 2,86 | –3,1 | | X | | | | | | |
| Gng2 | 2,98 | –3,4 | | | | | | | | X |
| Gpr162 | 4,13 | –4,3 | | | | | X | | | |
| H2-M3 | 2,70 | –3,2 | | | | | X | | | |
| Hfe | 5,45 | –3,5 | | | | | | | X | |
| Hip1 | 2,71 | –2,5 | | X | | | X | | | X |
| Hvcn1 | 3,48 | –3,4 | | | | | X | | | |
| Id3 | 3,61 | –3,4 | | | | | X | | | |
| Ifi203 | 2,64 | –3,0 | | | | X | X | | X | X |
| Ifi204 | 3,08 | –2,3 | | | | | | | | |
| Ifih1 | 4,04 | –2,6 | | | | | | | X | |

*Table 2 continued on next page*

*Table 2 continued*

| Gene symbol | -log(p-value) | Fold change (log2) | i-OCLs miRNAs (score 3–4) | | | | t-OCLs miRNAs (score 3–4) | | | |
| --- | --- | --- | --- | --- | --- | --- | --- | --- | --- | --- |
| | | | *Mir155* | *Mir146b* | *Mir342* | *Mir151* | *Mir185* | *Mir674* | *Mir26b* | *Mir29a* |
| Ifit3 | 2,82 | –2,9 | | | | X | X | | | |
| Ifngr1 | 2,38 | –2,1 | X | | | | | | | |
| Igf1 | 3,78 | –4,5 | X | | X | X | X | X | X | X |
| Igfbp4 | 3,21 | –4,8 | | | | | X | X | | |
| Ikbke | 3,62 | –3,9 | X | | | | X | | | X |
| Ikzf1 | 3,18 | –3,0 | | | | | X | X | | X |
| Il7r | 3,05 | –3,8 | X | | | | | | | |
| Irf9 | 2,46 | –2,4 | | | | | | X | | |
| Itgal | 2,94 | –3,9 | | | | | X | | X | |
| Kctd12 | 2,63 | –2,3 | | | X | X | | | | |
| Kif3a | 2,71 | –1,9 | X | | | | X | X | | |
| Klf2 | 2,62 | –2,7 | | | | X | | | | |
| Lamp2 | 2,45 | –2,4 | | X | | | | | | |
| Ldlrad3 | 2,71 | –3,8 | | | | | X | X | | X |
| Limd2 | 3,05 | –2,9 | | | | | X | X | | |
| Lipa | 2,53 | –2,4 | | X | | | | | X | |
| Lpar6 | 3,10 | –2,7 | | | | | X | | | |
| Lpcat2 | 2,57 | –3,7 | | | | | X | | | |
| Ltbp2 | 2,53 | –3,3 | | | X | | | | X | |
| Ly6e | 3,96 | –3,0 | | | | | | X | | |
| Ly9 | 2,34 | –1,9 | | | | | | X | | |
| Lyl1 | 3,48 | –3,9 | | | | | | X | | |
| Lyz2 | 3,62 | –4,0 | | | X | | | X | | |
| Maf | 3,59 | –4,0 | | | | X | | | | |
| Mafb | 2,41 | –3,4 | | | | | | | | X |
| Man2a2 | 3,96 | –2,9 | | X | | | | | | |
| Map3k1 | 2,42 | –2,0 | | | | | | | X | |
| Marcks | 2,80 | –2,2 | X | | | | | X | X | |
| Mef2c | 2,75 | –2,5 | X | | | | | X | X | |
| Mertk | 2,70 | –2,7 | | | | | | X | | |
| Mgll | 2,44 | –2,3 | | | | | X | | | |
| Mmp12 | 4,16 | –4,8 | | | | X | X | | | |
| Mpeg1 | 3,07 | –3,4 | X | | | | X | | | |
| Ms4a6b | 7,62 | –4,1 | | | | | | X | | |
| N4bp2l1 | 3,02 | –2,4 | | | | | X | | | X |
| Ncapg2 | 3,05 | –2,5 | X | X | | | X | X | | X |
| Ncf1 | 4,43 | –4,6 | | | | X | | X | | |
| Nek6 | 2,46 | –4,3 | | | | | | | X | |

*Table 2 continued on next page*

Table 2 continued

| Gene symbol | -log(p-value) | Fold change (log2) | i-OCLs miRNAs (score 3–4) | | | | t-OCLs miRNAs (score 3–4) | | | |
|---|---|---|---|---|---|---|---|---|---|---|
| | | | Mir155 | Mir146b | Mir342 | Mir151 | Mir185 | Mir674 | Mir26b | Mir29a |
| Neurl3 | 4,51 | –3,3 | | X | | | | | | |
| Nlrp3 | 3,29 | –4,2 | | | | | | X | | |
| Nrp1 | 2,36 | –2,5 | X | | | | | | | |
| Oas1a | 2,52 | –2,3 | | | X | | | | | |
| Osm | 3,07 | –3,2 | | | | | | | X | |
| P2rx7 | 3,53 | –3,1 | | | | | X | | | X |
| Pacs1 | 4,16 | –2,6 | | | | | X | | | |
| Parp14 | 2,93 | –2,4 | | | X | | X | | X | |
| Peli2 | 4,98 | –2,9 | | | | | | | X | X |
| Pgap1 | 3,22 | –2,5 | | | X | X | X | X | X | X |
| Pim1 | 2,93 | –2,7 | | | | | | X | | |
| Pld4 | 5,19 | –4,5 | | | | | | | | X |
| Plxnc1 | 4,24 | –4,1 | | | X | | | | | |
| Pmp22 | 2,98 | –3,0 | | | | | | | | X |
| Ppbp | 2,90 | 5,4 | | | | | | | X | |
| Ptgir | 3,30 | –3,2 | | | | | X | | | |
| Ptgs1 | 3,71 | –4,1 | | | | | | X | | |
| Rab3il1 | 3,19 | –3,5 | | | | | X | | | |
| Rab8b | 2,85 | –3,0 | X | | | | | | | |
| Rcsd1 | 3,74 | –3,7 | | | | | | X | | |
| Rnasel | 2,62 | –2,5 | | | | | | X | X | X |
| Rnf150 | 4,58 | –2,8 | | X | | X | X | X | X | X |
| Rsad2 | 2,53 | –3,0 | | X | X | X | | | | X |
| Rxra | 2,59 | –1,9 | | X | | | X | | | |
| S1pr1 | 3,26 | –3,6 | X | | | | X | | | |
| Sash1 | 3,01 | –2,6 | | | | X | | | | |
| Sash3 | 3,21 | –3,2 | | | | | X | | | |
| Sdc3 | 2,38 | –2,2 | | | X | X | X | X | | |
| Sepp1 | 3,53 | –4,2 | | | | X | | | | |
| Sgpp1 | 2,38 | –2,1 | | | | | X | X | | |
| Sidt2 | 2,64 | –2,1 | | | | | X | X | | X |
| Six1 | 3,60 | –5,0 | | | | X | X | X | | |
| Slamf7 | 4,94 | –4,2 | | | | | X | | | |
| Slamf8 | 3,05 | –2,2 | | | | | | | | X |
| Slc7a11 | 3,08 | –3,2 | | | | X | X | X | X | X |
| Slc9a3r1 | 2,94 | –2,6 | | X | | | | | | |
| Slfn5 | 4,24 | –2,7 | | | | | | X | | |
| Slfn8 | 2,76 | –3,5 | | | | | X | | | |

*Table 2 continued*

| Gene symbol | -log(p-value) | Fold change (log2) | i-OCLs miRNAs (score 3–4) | | | | t-OCLs miRNAs (score 3–4) | | | |
|---|---|---|---|---|---|---|---|---|---|---|
| | | | Mir155 | Mir146b | Mir342 | Mir151 | Mir185 | Mir674 | Mir26b | Mir29a |
| Snx24 | 2,99 | –3,2 | | X | | | X | X | | X |
| Snx30 | 2,79 | –3,2 | | | | | X | X | | |
| Stat2 | 2,76 | –1,8 | | X | | | | | | |
| Stom | 2,60 | –2,5 | | X | | X | | | | |
| Susd3 | 2,76 | –3,1 | | | | | | | | |
| Tanc2 | 2,65 | –3,1 | | X | X | X | X | X | X | |
| Tifa | 2,53 | –2,5 | X | | | | | | | |
| Tle3 | 4,70 | –3,7 | | | X | | X | | X | |
| Tlr13 | 4,27 | –4,1 | | | | | X | | | |
| Tmem154 | 3,16 | –5,6 | | X | X | | | X | | X |
| Tmem176a | 4,09 | –5,2 | | | | | X | | | |
| Tmem176b | 4,13 | –4,1 | | | | | X | | | |
| Tmem229b | 2,75 | –1,9 | X | | | | X | | | X |
| Tmem71 | 4,32 | –4,7 | | | | | | X | | |
| Tnfaip8l2 | 2,54 | –3,0 | | | | | X | | | |
| Tnfrsf26 | 3,50 | –4,4 | X | | | | X | | | X |
| Trim30a | 2,41 | –1,9 | X | | | | X | | | |
| Trim30d | 3,73 | –4,1 | | | | | X | X | | |
| Ugcg | 2,90 | –2,4 | | | | X | | | | |
| Usp18 | 4,07 | –3,0 | | | | | X | | | |
| Wls | 2,43 | –2,4 | | | | | | X | | |
| Ypel3 | 2,39 | –1,6 | | | | | X | | | |
| Zcchc24 | 3,30 | –3,2 | | X | | | X | | X | |
| Zfhx3 | 3,46 | –2,4 | | | | | X | | X | X |
| Zfp36l1 | 2,36 | –2,1 | | | | | X | | | X |
| Zfp608 | 2,38 | –2,6 | | | | | X | | X | |

*Britton et al., 2014*; *Lawenius et al., 2022*) or to a site-specific response to yeast derivates. It had no effect on the number of osteoblasts and osteocytes in vivo (*Figure 2—figure supplement 2a–d*) and on the mineralization capacity of osteoblasts in vitro (*Figure 2—figure supplement 2e*).

We then checked whether *Sb* treatment affects inflammatory parameters and microbiota metabolites that are known to influence bone remodeling (*Zaiss et al., 2019*). Evaluation of the gut barrier integrity by fluorescein isothiocyanate (FITC)-dextran assay showed a reduction in serum dextran concentration in *Sb*-treated OVX mice to the level observed in SHAM mice (*Figure 3a*) confirming the protective effect of *Sb* on the intestinal barrier as already reported in other pathological contexts (*Terciolo et al., 2019*). Modifications in the gut microbiome were evaluated by dosage of serum concentrations of metabolites produced by commensal bacteria such as propionate and butyrate, two major short chain fatty acids (SCFA) as well as lactate produced by lactic acid bacteria (LAB). Serum propionate, butyrate, and lactate were reduced in OVX mice, and treatment with *Sb* reversed this decrease, although not reaching statistical significance for lactate and butyrate (*Figure 3b–d*).

Interestingly, in *Sb*-treated OVX mice, we also observed a decreased proportion of BM CD4[+] Tnf-α-producing T cells that have been reported to be responsible for increased osteoclastogenesis

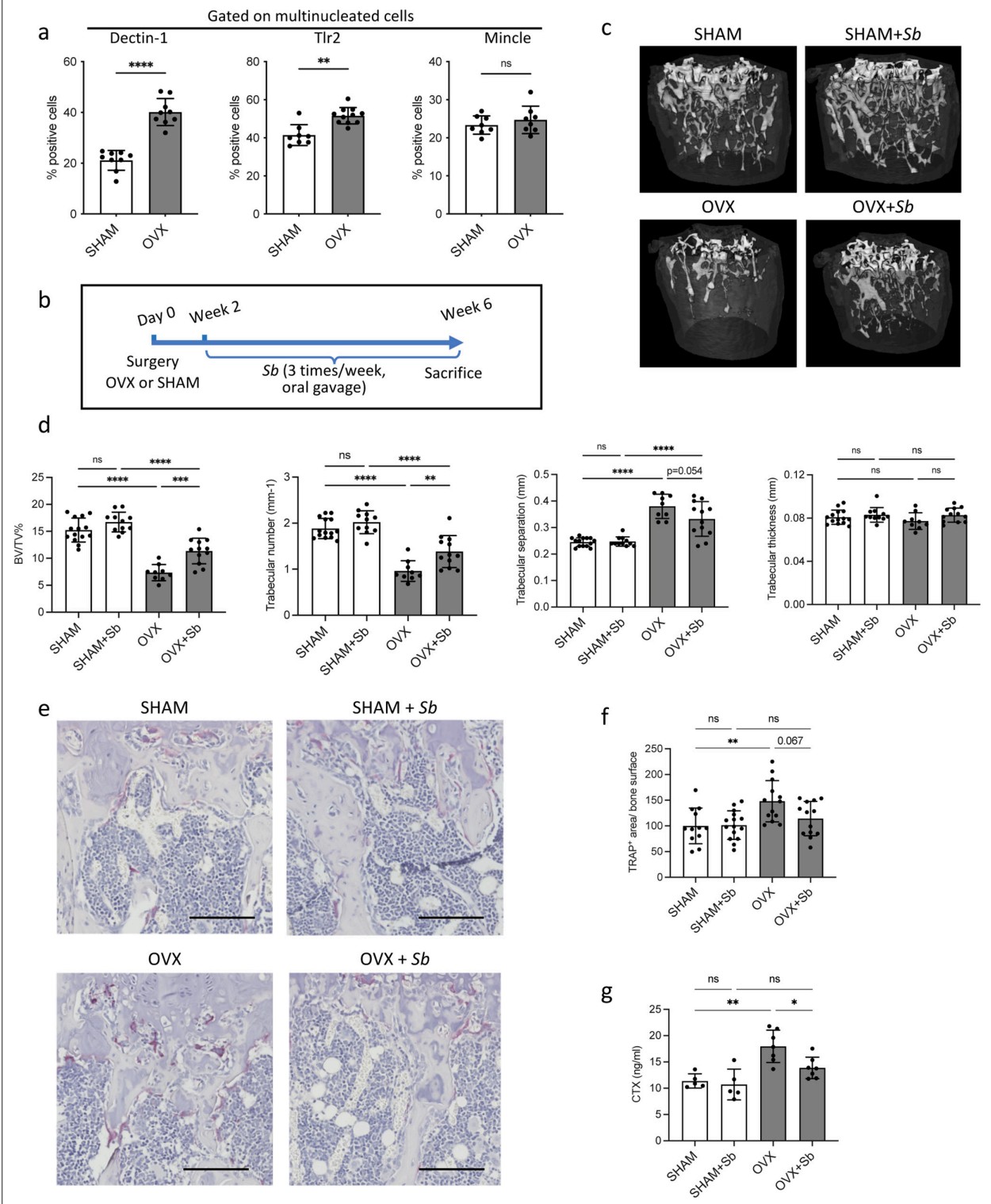

**Figure 2.** The probiotic yeast *Saccharomyces boulardii* (*Sb*) has beneficial effects on bone loss in osteoporosis. (**a**) Quantification (mean ± SD) of FACS analysis of Dectin-1+, Tlr2+, and Mincle+ mature osteoclasts (OCLs; ≥three nuclei, gated as shown in *Figure 1—figure supplement 1b*, differentiated from the bone marrow of SHAM and ovariectomized (OVX)) mice, 6 weeks after surgery. (**b**) Schematic representation of the experimental procedure. Uterus and body weight were measured to control the quality of ovariectomy (*Figure 2—figure supplement 1a–b*). (**c**) Representative microcomputed tomography images of femurs from SHAM and OVX mice ± *Sb* administration. (**d**) Histograms indicate mean ± SD of trabecular bone volume fraction, trabecular number, separation, and thickness. Cortical parameters were also measured (*Figure 2—figure supplement 1c*). (**e**) Histological analysis of

*Figure 2 continued on next page*

*Figure 2 continued*

OCLs using TRAcP staining (in purple) on tibias from SHAM and OVX mice treated or not with *Sb*. Scale bars: 100 µm. (**f**) Histogram indicates the mean ± SD of TRAcP+ area per bone surface for each condition. Three images of four to five biological replicates were analyzed. Osterix+ and Sost+ cells were also analyzed (*Figure 2—figure supplement 2a–d*). The mineralization capacity of osteoblasts was also analyzed in vitro in the presence of *Sb*-conditioned medium (*Figure 2—figure supplement 2e*). (**g**) Serum cross-linked C-telopeptides of type I collagen (CTX) were measured by ELISA (n≥5 biological replicates per condition). *p<0.05; **p<0.01; ***p<0.001; ****p<0.0001; ns, non-significant differences.

The online version of this article includes the following figure supplement(s) for figure 2:

**Figure supplement 1.** Cortical bone parameters of SHAM and OVX mice.

**Figure supplement 2.** Effect of *Sb* treatment on osteoblasts and osteocytes.

in OVX mice (*Cenci et al., 2000*; *Figure 3e*). Lastly, we evaluated i-OCL differentiation by analyzing mature BM-derived OCLs for their expression of Cx3cr1, the previously described marker for i-OCLs (*Ibáñez et al., 2016*). In *Sb*-treated OVX mice, the proportion of Cx3cr1+ i-OCLs was significantly reduced compared to non-treated OVX mice, while there was no alteration of Cx3cr1+ OCLs in SHAM mice (*Figure 3f*).

## Stimulation of TLRs and CLRs influences i-OCL differentiation

Our results showed that the beneficial effect of *Sb* in OVX mice was in part due to a systemic effect on inflammatory parameters responsible for pathological osteoclastogenesis. However, yeast derivatives such as ß-glucans are known to translocate to the blood and organs through the gut barrier (*Isnard et al., 2021*; *Rice et al., 2005*) and can therefore directly affect cells expressing PRR that recognize

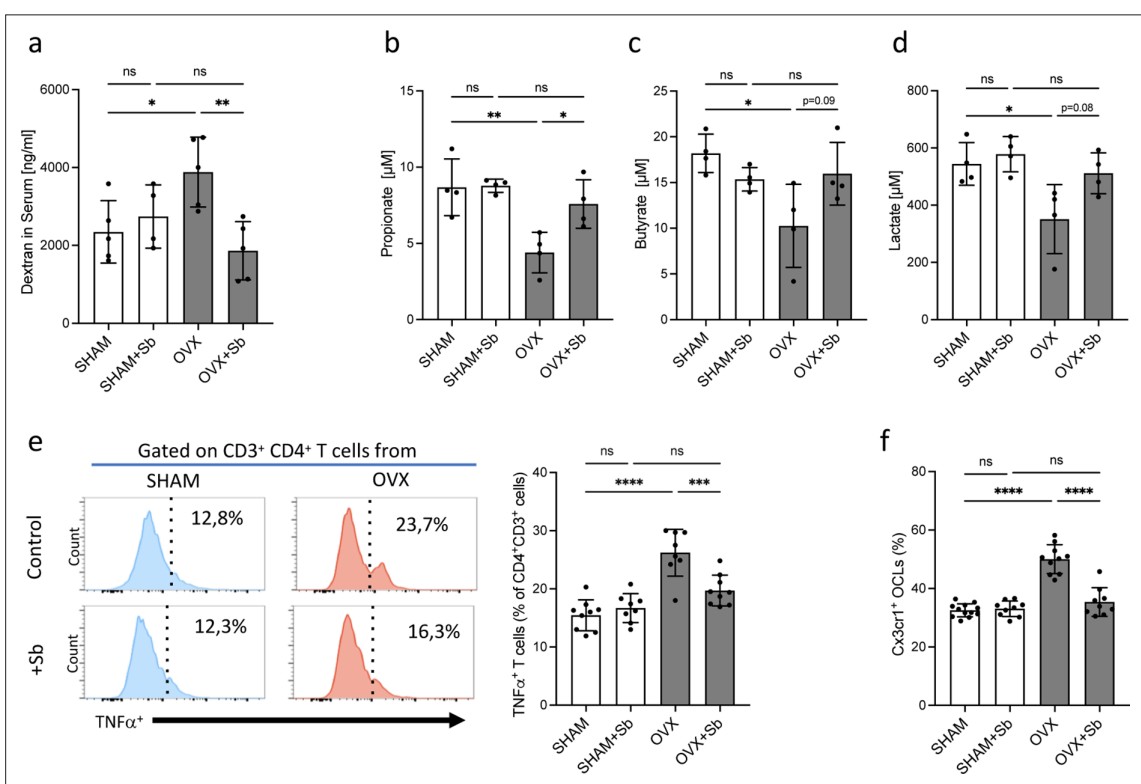

**Figure 3.** *Saccharomyces boulardii (Sb)* reduces inflammatory parameters in osteoporosis. (**a**) Integrity of the intestinal barrier permeability was analyzed by fluorometry on the serum of mice that received oral gavage of dextran-fluorescein isothiocyanate 1 hr before sacrifice (n=4–5 biological replicates) (mean ± SD). (**b–d**) Concentrations (mean ± SD) of (**b**) propionate, (**c**) butyrate, and (**d**) lactate were measured by chromatography in the serum of SHAM and ovariectomized (OVX) mice treated or not with Sb (n=4 biological replicate per group). *p<0.05; **p<0.01. (**e**) FACS analysis of Tnf-α-producing CD4+ T cells in the bone marrow (BM) of OVX and SHAM control mice with or without *Sb* treatment (n=8–10 biological replicate per group). (**f**) Proportion of mature Cx3cr1+ BM-derived OCLs (≥three nuclei, gated as shown in *Figure 1—figure supplement 1b*) from SHAM and OVX mice treated or not with *Sb* (n=10–14 biological replicate per group) was determined by FACS analysis. *p<0.05; **p<0.01; ***p<0.001; ****p<0.0001; ns, non-significant differences.

them. Thus, we determined the direct effect of agonists of these receptors on the differentiation of t-OCLs and i-OCLs. We used curdlan and zymosan as Dectin-1 and Tlr2 agonists and glucosyl-6-tetradecyloctadecanoate (GlcC$_{14}$C$_{18}$, a synthetic C6-branched glycolipid) to stimulate Mincle. The formation of t-OCLs from BM CD11b$^+$ cells was not affected by any of these agonists (*Figure 4a–c*), consistent with their low expression of Dectin-1, Tlr2, and Mincle receptors (*Figure 1k–l*). In contrast, curdlan, zymosan, and GlcC$_{14}$C$_{18}$ dramatically inhibited the differentiation of BM-derived DCs into i-OCLs (*Figure 4a–c*). This effect was not due to an increased apoptosis of the cells in the presence of the agonists (*Figure 4—figure supplement 1a–d*). To confirm the involvement of Dectin-1, Tlr2, and Mincle in these effects, we used neutralizing antibodies and siRNA. Anti-Dectin-1 and anti-Mincle antibodies reversed the inhibitory effect of curdlan and GlcC$_{14}$C$_{18}$, respectively, on i-OCL formation compared to the control isotype (*Figure 4d–e*). *Tlr2* siRNA decreased the expression level of membrane Tlr2, as expected (*Figure 4—figure supplement 1e*) and was able to abrogate the decrease of i-OCL differentiation induced by curdlan but not by zymosan (*Figure 4f*).

These data revealed that Dectin-1, Tlr2, and Mincle activation specifically inhibits the formation of i-OCLs without affecting t-OCLs. Additionally, we performed in vitro assays to assess the effect of these agonists on mature i-OCL activity. Our results revealed that, in addition to blocking i-OCL differentiation, agonists of these receptors also reduced their capacity to degrade mineralized matrix (*Figure 4g*), whereas they do not affect the activity of t-OCLs (*Figure 4—figure supplement 1f*).

In line with the aforementioned results, the Dectin-1 and Tlr2 agonists (curdlan and zymosan) also strongly reduced the differentiation of OCLs derived from BM cells of OVX mice in vitro, while they had no significant effect on the differentiation of OCLs generated from SHAM control mice (*Figure 4h*). However, GlcC$_{14}$C$_{18}$ reduced OCL differentiation of progenitors from OVX mice but also from SHAM mice (*Figure 4h*), according to the equivalent expression level of Mincle in these cells (*Figure 2a*).

The inhibition of the differentiation of i-OCL by a Mincle agonist was contrasting with a recent publication showing that *Clec4e$^{-/-}$* mice deficient in Mincle have increased bone mass and that sensing of necrotic osteocytes by Mincle increases OCL differentiation (*Andreev et al., 2020*). Accordingly, using µCT analysis, we observed the same increase in BV/TV, trabecular number, and trabecular thickness in *Clec4e* KO mice compared to controls, as already demonstrated (*Andreev et al., 2020*; *Figure 4—figure supplement 2a*). However, in vitro assays revealed that BM cells, BM-DCs, and BM-MNs from *Clec4e$^{-/-}$* mice differentiate into OCLs at least as efficiently as cells from controls (*Figure 4—figure supplement 2b*), contrasting with data from the literature (*Andreev et al., 2020*), which could be explained by different protocols used for in vitro OCL differentiation. Moreover, inhibition of i-OCL differentiation by GlcC$_{14}$C$_{18}$ was decreased in cells from *Clec4e$^{-/-}$* compared to *Clec4e$^{+/+}$* mice confirming that Mincle is involved in this inhibition (*Figure 4—figure supplement 2c*). These data revealed that implication of Mincle in OCL differentiation is much more complex than expected.

Downstream signaling of Dectin-1 and Mincle largely involves activation of the spleen tyrosine kinase, Syk, which is also required for the differentiation and activity of OCLs (*Mócsai et al., 2004*). Accordingly, we found that BM-derived DCs from *CD11cΔSyk* mice that have selective depletion of *Syk* in CD11c$^+$ cells (*Iborra et al., 2012*) failed to differentiate into OCLs (*Figure 5a*), indicating that *Syk* expression by DCs is required for i-OCL formation. As expected, FACS analysis showed that addition of curdlan to the OCL differentiation medium rapidly induced Syk phosphorylation (*Figure 5b*), which was followed at 24 hr by an increased proportion of MHC-II$^+$ and CD80$^+$CD86$^+$ DCs, revealing the maturation of BM-derived DCs (*Figure 5c*). This treatment simultaneously reduced the proportion of RANK$^+$ cells expressing Csf1r (CD115) and FcgRII/III (CD16/32), all required for OCL differentiation (*Figure 5d*). Furthermore, it also downregulated the expression of *Syk*, as well as its downstream targets *Nfatc1* and *Ctsk* (*Figure 5e*). These results show that despite a rapid activation of Syk in BM-DCs treated with M-csf and Rank-L upon addition of curdlan, *Syk* expression decreases with time, as previously shown (*Yamasaki et al., 2014*), as well as the capacity of BM-DCs to differentiate into i-OCLs.

To investigate whether yeast probiotics have the same effect as the PRR agonists on the differentiation of i-OCLs, we used *Sb*-conditioned medium (*Sb*-CM). *Sb*-CM completely blocked the differentiation of i-OCL progenitors while it did not affect the formation of t-OCLs, except at very high concentrations (*Figure 6a*). This inhibitory effect was not associated with an increase in cell apoptosis (*Figure 6b*). Moreover, *Sb*-CM also reduced the resorption capacity of OCLs in vitro (*Figure 6c*). Next, we addressed the involvement of Dectin-1, Tlr2, and Mincle in the inhibitory effect of *Sb*-CM on

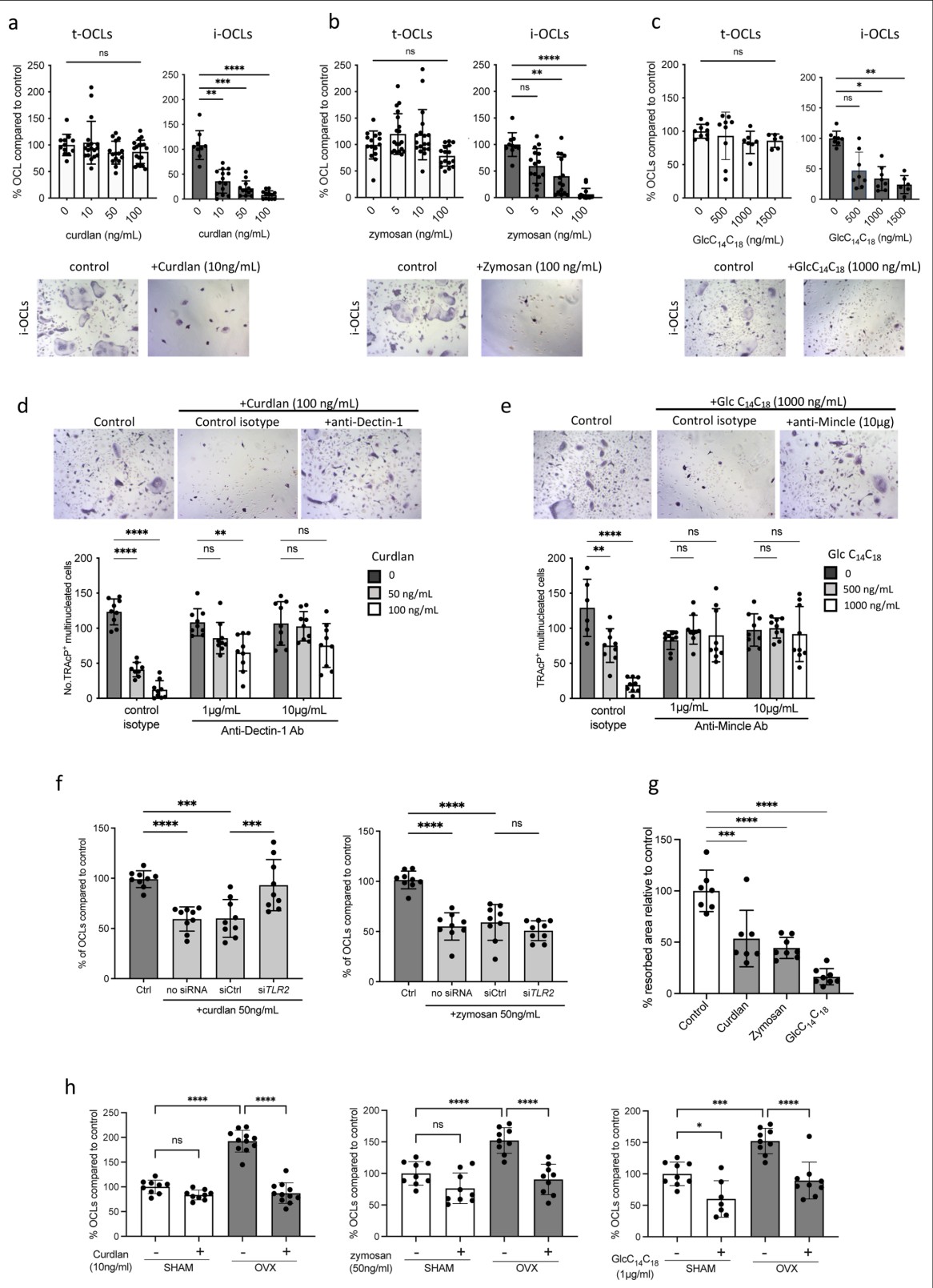

**Figure 4.** Implication of Tlr2, Dectin-1, and Mincle in the differentiation of inflammatory osteoclasts (i-OCLs). (**a–c**) Quantification of the differentiation of tolerogenic OCLs (t-OCLs) and i-OCLs in the presence of indicated concentrations of curdlan, zymosan, and GlcC$_{14}$C$_{18}$. Upper panels: TRAcP$^+$ cells with three or more nuclei were counted as OCLs. Histogram indicates the mean ± SD of the OCL normalized to the control condition (0 ng/mL agonist). Bottom panels: representative image of TRAcP staining for the control (without agonist) and agonist-treated i-OCLs at the indicated concentration.

*Figure 4 continued on next page*

*Figure 4 continued*

Quantification of cells viability in the presence of the agonists was also performed by FACS analysis (*Figure 4—figure supplement 1a–d*). (**d-e**) TRAcP staining and quantification of OCLs (TRAcP$^+$ cells ≥three nuclei) differentiated in the presence of control isotypes or blocking antibodies against (**d**) Dectin-1 and (**e**) Mincle. (**f**) Quantification of i-OCLs differentiated in the presence of control siRNA or siRNA targeting *Tlr2*. Level of Tlr2 expression in the presence and absence of siRNAs was evaluated by FACS (*Figure 4—figure supplement 1e*). (**g**) OCL activity was evaluated by seeding bone marrow (BM)-derived dendritic cells on plates coated with resorbable matrix. After i-OCLs started to fuse (day 3 of differentiation), the different agonists were added at the following concentration: curdlan, 50 ng/mL; zymosan, 50 ng/mL, GlcC$_{14}$C$_{18}$, 1000 ng/mL. Resorbed area was quantified after 3 days. Evaluation of the effect of the agonists on t-OCL activity was also analyzed (*Figure 4—figure supplement 1f*). Specific analysis of the effect of Mincle stimulation on OCL formation was assessed using *Clec4e*-KO mice (*Figure 4—figure supplement 2*). (**h**) Enumeration of in vitro differentiated OCLs from BM cells of OVX and SHAM mice in the presence of indicated concentrations of curdlan, zymosan, and GlcC$_{14}$C$_{18}$ (Glc). TRAcP$^+$ cells with three or more nuclei were counted as OCLs (n=8–11). *p<0.05; **p<0.01; ***p<0.001; ****p<0.0001; ns, non-significant differences.

The online version of this article includes the following figure supplement(s) for figure 4:

**Figure supplement 1.** Effects of agonists on cell viability and t-OCL activity.

**Figure supplement 2.** Bone phenotype and OCL differentiation in *Clec4e* deficient mice.

i-OCLs as described above. While *Tlr2*-siRNA had no effect (*Figure 6d*), anti-Dectin-1 and anti-Mincle blocking antibodies completely abrogated the inhibitory effect of *Sb*-CM on the differentiation of i-OCLs, demonstrating the prominent role of these receptors in mediating the effect of *Sb* on inflammatory osteoclastogenesis (*Figure 6e*). Moreover, as curdlan, *Sb*-CM strongly stimulated BM-DC maturation (*Figure 6f*) while it dramatically decreased the proportion of Rank$^+$ Csf1r$^+$ (CD115$^+$) and FcgRII/III$^+$ (CD16/32$^+$) cells representing OCL progenitors (*Figure 6g*). These results revealed that derivatives from *Sb* recognized by PRRs, such as ß-glucans, interfere with the capacity of BM-derived DCs to differentiate into i-OCLs and with the activity of these OCLs.

## Discussion

The present study demonstrates that i-OCLs and t-OCLs are two distinct OCL populations with specific molecular signatures that differ in their preferential use of differentiation pathways and their capacity to sense the environment through the PRRs Dectin-1, Tlr2, and Mincle. Based on these specificities, we could show that targeting these receptors reduces bone loss in OVX mice and inhibits specifically the differentiation of i-OCLs while sparing t-OCLs.

OCL heterogeneity has been neglected for a long time, and the mechanisms regulating the emergence and function of i-OCLs are still largely unknown. Our transcriptomic profiling shows that, while being comparable in their expression of bone resorption-associated genes, t-OCLs and i-OCLs differ in their expression of genes associated with immune responses, which provides additional evidence for their previously reported divergent immune properties (*Ibáñez et al., 2016*). It also highlights differences in genes related to OCL differentiation. Osteoclastogenesis is regulated by two main pathways, the classical Rank-associated and the co-stimulatory Ig-like receptor-associated pathways, both of which are required to promote OCL differentiation (*Humphrey and Nakamura, 2016*; *Koga et al., 2004*; *Seeling et al., 2013*). Increased Rank-L levels as well as stimulation of FcγR and ITAM signaling are involved in pathological bone resorption in rheumatic diseases such as rheumatoid arthritis (*Herman et al., 2008*; *Ochi et al., 2007*). However, to date these pathways have not been described to be preferentially used by specific populations of OCLs. Here, we show that the implication of the co-stimulatory pathway in diseases associated with high bone resorption could be linked to its upregulation during i-OCL differentiation, which suggests that specific targeting of this pathway and its related molecules could limit inflammatory bone loss with a minimal impact on OCLs involved in physiological bone remodeling.

Remarkably, Tlr2, Dectin-1, and Mincle are increased in i-OCLs and OCLs from OVX mice compared to t-OCLs and to those from SHAM mice. These receptors are expressed in myeloid cells and sense microbial structures from bacteria, fungi, or parasites, including lipoproteins, glycoproteins, peptidoglycan for Tlr2, β-glucans for Dectin-1 and Tlr2, and α-mannose for Mincle (*Sancho and Reis e Sousa, 2012*; *Savva and Roger, 2013*). Moreover, they interact with each other and with Fcγ receptors for the induction of inflammatory responses (*Gantner et al., 2003*; *Sato et al., 2003*). Very few studies have investigated the effect of microbial β-glucans and α-mannose on osteoclastogenesis.

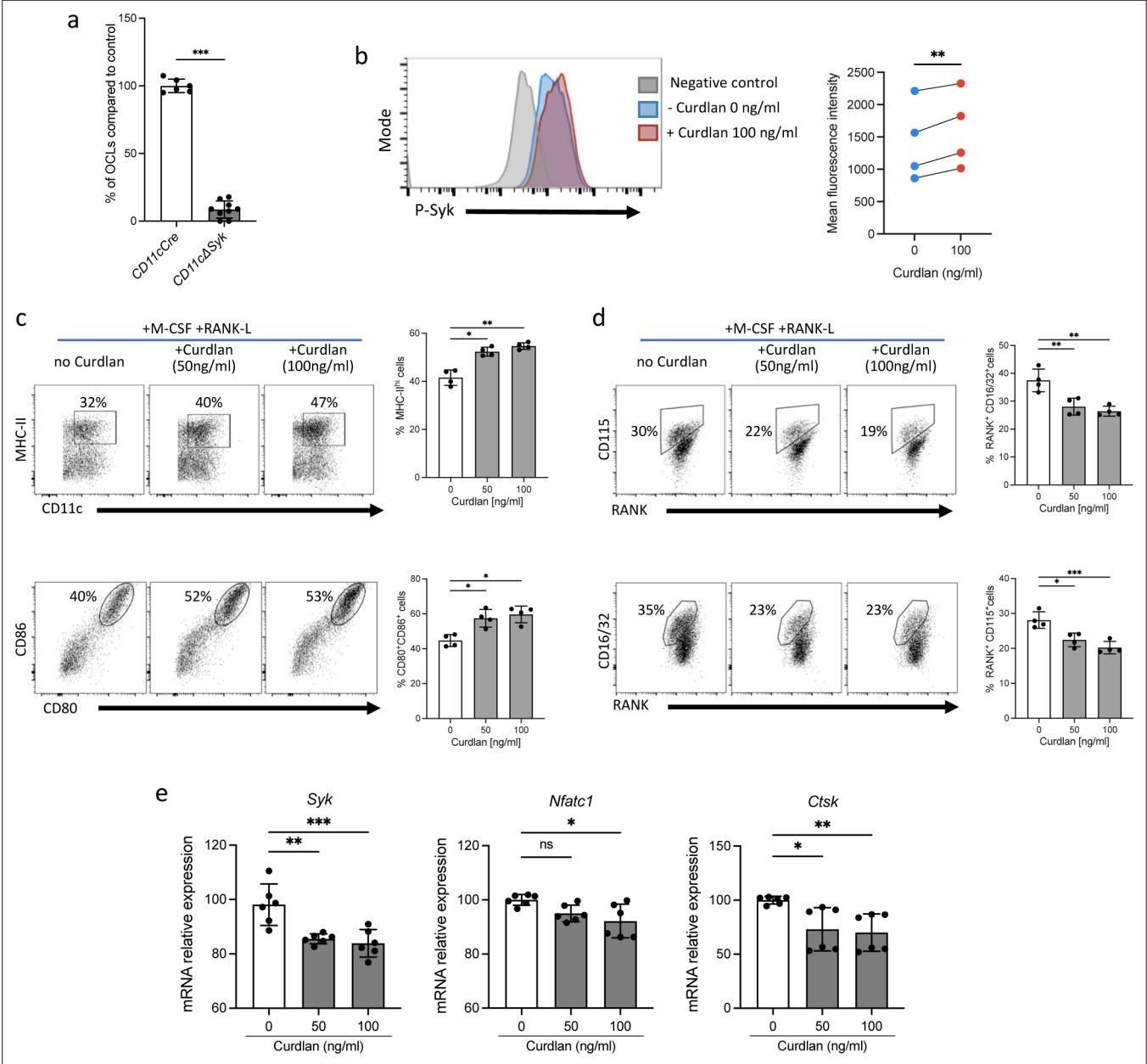

**Figure 5.** Curdlan alters the capacity of dendritic cells (DCs) to differentiate into inflammatory osteoclasts (OCLs). (**a**) Quantification of differentiation of inflammatory OCLs from *CD11cΔSyk* and control mice (n=9 and 6 biological replicates, respectively). (**b**) Flow cytometry analysis of spleen tyrosine kinase (Syk) phosphorylation after 15 min of stimulation with 100 ng/mL of curdlan on bone marrow (BM)-derived DCs cultured. The right panel shows the increase in the mean fluorescence intensity revealing increased Syk phophorylation (n=4 biological replicates). (**c–d**) FACS plots and quantification (mean ± SD) of the expression of (**c**) MHC-II, CD80, and CD86; and (**d**) CD115 (Csfr1), Rank, and CD16/32 (FcgrII/III) on BM-DCs (n=4 biological replicates per group) cultured in osteoclast differentiation medium and stimulated or not for 24 hr with the indicated curdlan concentrations (**e**) Quantitative real-time PCR analysis of the expression of *Syk, Nfatc1,* and *Ctsk* on BM-DCs cultured in osteoclast differentiation medium and stimulated or not for 72 hr with the indicated curdlan concentrations. *p<0.05; **p<0.01; ***p<0.001; ****p<0.0001; ns, non-significant differences.

Tlr2 stimulation was shown to reduce osteoclastogenesis from non-committed OCL progenitors but to increase OCL survival and differentiation of Rank-L-primed pre-OCLs (*Souza and Lerner, 2019*; *Takami et al., 2002*). High concentrations of curdlan (in the range of μg/mL) have been previously reported to decrease the overall differentiation of OCLs from Dectin-1-transfected RAW cells or from BM cells by upregulating *Mafb* and reducing *Nfatc1* expression (*Zhu et al., 2017*). Our results extend these data by revealing that with much lower doses of curdlan (range of ng/mL), only i-OCL formation is inhibited whereas other OCLs are not affected, in line with the huge difference in Dectin-1 and

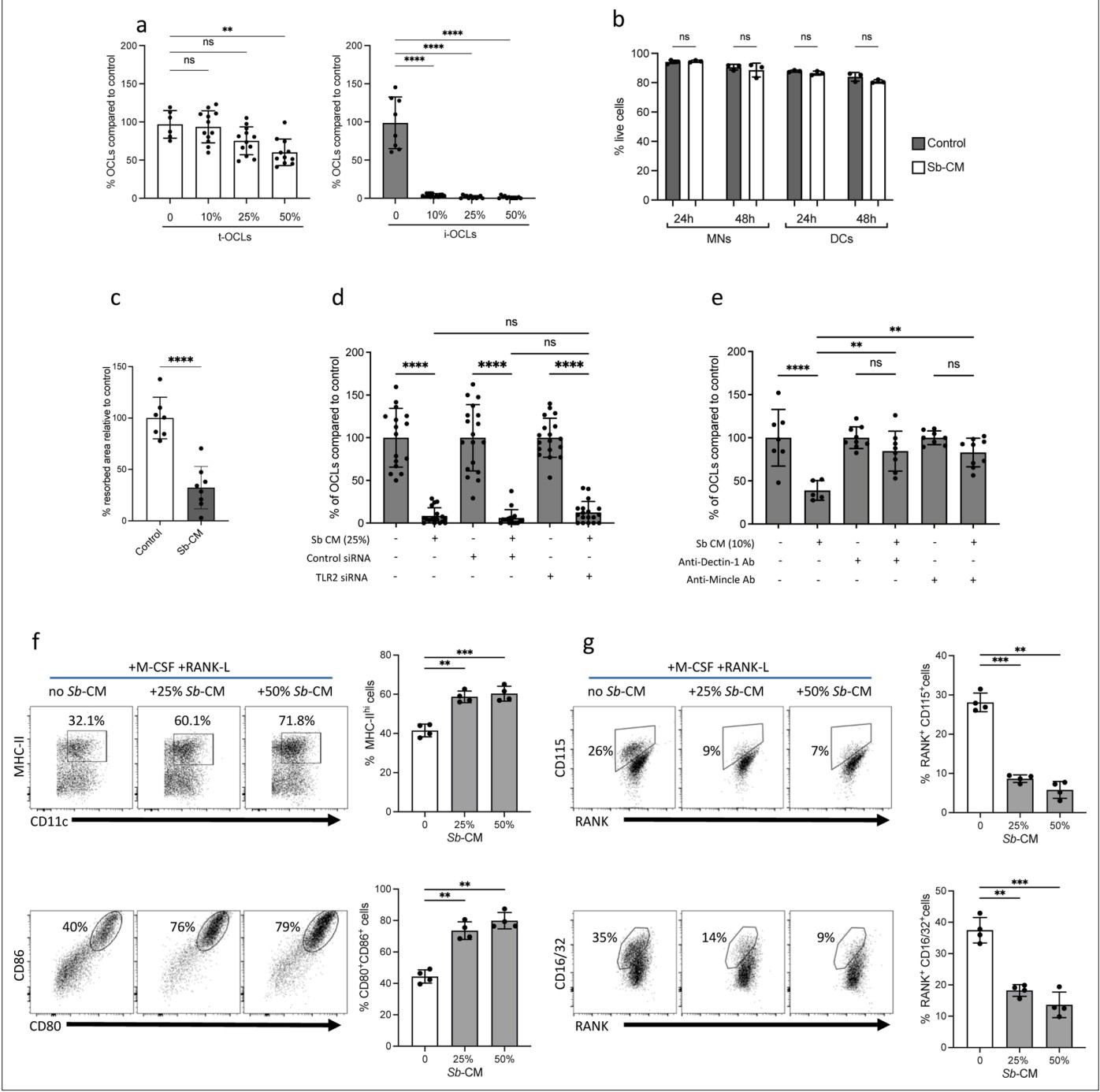

**Figure 6.** The probiotic yeast *Saccharomyces boulardii* (*Sb*) modulates the differentiation of inflammatory osteoclasts (i-OCLs). (**a**) Quantification of differentiated tolerogenic OCLs (t-OCLs) and i-OCLs in the presence of indicated concentrations of *Sb*-conditioned medium (*Sb*-CM). (**b**) Apotosis of OCL progenitors (monocyte cells [MNs] and dendritic cells [DCs]) cultured in OCL differentiation medium in the absence (Control) or presence of *Sb*-CM (25%) was measured by FACS. Histograms indicate mean ± SD of live cells (annexin-V$^{neg}$ propidium iodide$^{neg}$) after 24 hr and 48 hr. (**c**) OCL activity was evaluated by on plates coated with resorbable matrix by adding *Sb*-CM (25%) when OCL fusion is starting (day 3 of differentiation). Resorbed area was quantified 3 days after addition of *Sb*-CM. Control panels are the same as in *Figure 4g* because the experiments were performed together. (**d**) Differentiation of i-OCLs was performed in the presence of control siRNA or siRNA, targeting *Tlr2* in the absence or presence of *Sb*-CM. (**e**) Differentiation of i-OCLs was performed in the presence of blocking antibodies for Dectin-1 and Mincle in the absence or presence of *Sb*-CM. For all experiments, TRAcP$^+$ multinucleated cells with ≥three nuclei were considered as OCLs and enumerated after 6 days of culture. (**f–g**) FACS plots and quantification (mean ± SD) of the expression of (**f**) MHC-II, CD80, and CD86; and (**g**) CD115 (Csfr1), Rank, and CD16/32 (FcgrII/III) on BM-DCs (n=4 biological replicates) cultured in osteoclast differentiation medium and stimulated or not for 24 hr with the indicated % of *Sb*-CM. The control panels

*Figure 6 continued on next page*

*Figure 6 continued*

(without *Sb*) are identical to those of **Figure 5c–d** as the experiments were performed together. *p<0.05; **p<0.01; ***p<0.001; ****p<0.0001; ns, non-significant differences.

Tlr2 expression by their respective progenitors. Interestingly, here we show that the yeast probiotic *Sb* significantly reduces bone loss in OVX mice. *Sb* has proven its beneficial probiotic effects in several gastrointestinal disorders. It favors regeneration of the integrity of the epithelial barrier in colitis and restores the gut microbiota after antibiotic treatment or diarrhea by increasing SCFA-producing bacteria such as *Lachnospiraceae, Bacteroides, Ruminococcus,* and *Prevotellaceae* (**Moré and Swidsinski, 2015**; **Terciolo et al., 2019**; **Yu et al., 2017**). In line with this, our results demonstrate that *Sb* improves the gut barrier function in the context of OVX-induced osteoporosis. It also normalizes the concentration of SCFAs and lactate in OVX mice, which strongly suggests that *Sb* favors bacteria species producing these metabolites. Indeed previous studies reported that the diversity of intestinal microbiota is lower in OVX mice than in SHAM mice, and its composition is altered (**Li et al., 2021**; **Zaiss et al., 2019**). LAB as well as SCFA-producing bacteria have a beneficial effect on bone (**Zaiss et al., 2019**) which probably participates in the reduced bone loss observed in *Sb*-treated OVX mice. Moreover, gut dysbiosis and increased intestinal permeability reported in osteoporosis (**He et al., 2020**; **Xu et al., 2020**) induce activation of osteoclastogenic CD4+ T cells over-producing Rank-L and Tnf-α (**D'Amelio et al., 2008**; **Li et al., 2016**). Thus, the reduction in BM Tnf-α-producing CD4+ T cells observed after *Sb* treatment in OVX mice is likely due to its beneficial effect on the gut, as described for bacterial probiotics (**Li et al., 2016**). As Tnf-α+ CD4+ T cells are major inducers of i-OCLs (**Ciucci et al., 2015**; **Ibáñez et al., 2016**; **Madel et al., 2019**), this effect probably also participates in the decreased proportion of i-OCLs observed in *Sb*-treated OVX mice. In addition, in various models of gut disorders, *Sb* has been described to increase the production of anti-inflammatory cytokines such as IL-10 and IL-4 (**Czerucka and Rampal, 2019**; **Pais et al., 2020**) that are potent inhibitors of OCL formation (**Fujii et al., 2012**; **Park-Min et al., 2009**). Despite we did not detect significant difference in serum IL-10 concentration after Sb treatment in OVX mice (data not shown), we could not rule out local cytokine modification in the BM.

In addition to its systemic effects on inflammation and microbiota, *Sb* is also likely to directly affect i-OCL differentiation. The yeast cell wall components, ß-glucans, are well known to cross the gut barrier and translocate to the blood from where they can disseminate to organs (**Isnard et al., 2021**; **Rice et al., 2005**) and can directly modulate cells from the monocytic family (**Ibáñez et al., 2019a**), which includes circulating OCL progenitors, through stimulation of CLRs. Consequently, in vivo they can possibly exert the same direct inhibitory effect on the differentiation of i-OCLs as observed in vitro. This is confirmed by our in vitro analysis showing a specific inhibition of i-OCL differentiation compared to t-OCLs by *Sb*, as well as low doses of agonists of Tlr2, Dectin-1, and Mincle, which is in line with the much higher expression of these receptors in BM-derived DCs than in BM MNs.

The inhibitory effect of Mincle stimulation observed here is contrasting with a recent publication reporting that sensing of necrotic cells through Mincle by OCL progenitors stimulates OCL formation (**Andreev et al., 2020**). *Clec4e* KO mice have increased bone mass (**Andreev et al., 2020**), a result confirmed by our study. However, we also observed a similar or even higher OCL formation from progenitors from *Clec4e* KO compared to control mice. Mincle is not expressed by osteoblasts (**Andreev et al., 2020**), but the increased bone mass observed in *Clec4e* KO mice could be related to indirect effects of Mincle on osteoblasts and OCLs through myeloid cells that are well-known regulators of bone cell differentiation and activity (**Yahara et al., 2021**). Moreover, Mincle KO mice display reduced CD4+IL-17+ T cells (**Martínez-López et al., 2019**), which may also contribute to a reduced OCL differentiation and increased bone mass. It is therefore likely that depending on whether Mincle is stimulated by exogenous microbial signals (as in our study) or endogenous necrotic signals (as in Andreev's study), this receptor has divergent effects on osteoclastogenesis. These observations revealed the complex effects of Mincle on osteoclastogenesis, as already reported for other monocytic cells (**Patin et al., 2017**).

The kinase Syk plays a major role in Dectin-1 and Mincle signaling pathways, and indeed, Syk is rapidly phosphorylated upon curdlan stimulation in BM-DCs. On the other hand, Syk is required for efficient OCL differentiation (**Zou et al., 2007**), including from DCs as shown here. However, in DCs stimulated with curdlan in osteoclastogenic medium, the expression of *Syk* decreases with time

together with *Nfatc1*, a master gene of OCL differentiation, and *Ctsk*, a main marker of OCLs required for their activity, which participates in reducing i-OCL formation. These results are in agreement with the literature showing in the same conditions a degradation of Syk after treatment with high doses of curdlan (*Yamasaki et al., 2014*). In DCs, the interaction between Tlr2 and Dectin-1 is involved in the stimulation of NF-κB and in the production of inflammatory cytokines such as Tnf-α and IL-12 (*Gantner et al., 2003*; *Sancho and Reis e Sousa, 2012*). This strongly induces DC activation and therefore potent phagocytic and anti-microbial responses (*Gantner et al., 2003*; *Sancho and Reis e Sousa, 2012*). Accordingly, we show that treatment with curdlan or with *Sb*-CM induces DC maturation while it decreases the proportion of Rank⁺ cells expressing Csf1r and FcgrII/III, all of which are required for OCL formation. Thus, stimulation of DCs with the PRR agonists has a dual effect, i.e., increasing their maturation and simultaneously reducing their ability to give rise to i-OCLs.

In conclusion, we here demonstrated that i-OCLs differ from t-OCLs in the control of their differentiation and in their capacity to sense their environment and in particular to respond to stimuli through CLR and TLR activation. Based on these properties, we demonstrated that the probiotic yeast *Sb* has a beneficial effect on bone loss in osteoporotic mice by restoring gut barrier integrity and reducing osteoclastogenic CD4⁺ T cells thereby reducing indirectly i-OCLs. Moreover, it also directly targets i-OCL progenitors to interfere with their differentiation.

These insights open new interesting perspectives for the treatment of pathological bone resorption by demonstrating that the Ig-like receptor costimulatory pathway and the associated PRR pathway, rather than the Rank pathway, could represent efficient therapeutic targets to specifically impact on inflammatory osteoclastogenesis while maintaining physiological OCL resorption. The dose of *Sb* used in mice is much higher than that recommended for humans. However, this is also true in studies evaluating probiotic lactobacillus strains in osteoporosis that have shown protective effects at much lower doses in humans than in mice (*Britton et al., 2014*; *Nilsson et al., 2018*; *Jansson et al., 2019*). Therefore, *Sb* administration in osteoporotic patients would represent a feasible possibility. These novel insights and regulatory mechanisms mediated by yeast probiotics could provide new therapeutic options to overcome the global inhibition of OCLs and the resulting impaired bone quality associated with current anti-resorptive therapies.

# Materials and methods

**Key resources table**

| Reagent type (species) or resource | Designation | Source or reference | Identifiers | Additional information |
|---|---|---|---|---|
| Strain, strain background (*Mus musculus*) | *CD11cΔSyk* and *CD11c-Cre* | **Iborra et al., 2012** | | |
| Strain, strain background (*M. musculus*) | B6.Cg-*Clec4e*^tm1.1Cfg | **Wells et al., 2008** | *Clec4e*⁻ᐟ⁻ | |
| Strain, strain background (*yeast*) | *Saccharomyces boulardii* (Ultralevure) | Biocodex | | |
| Antibody | CD11b | ThermoFisher Scientific | clone M1/70 | 1:100 |
| Antibody | CD11c | BD Biosciences | clone HL3 | 1:200, for cell isolation |
| Antibody | CD11c | eBioscience | clone N418 | 1:200 |
| Antibody | Dectin-1 | ThermoFisher Scientific | clone bg1fpj | 1:100 |
| Antibody | Dectin-1 | ThermoFisher Scientific | clone bg1fpj | 1:100 |
| Antibody | Dectin-2 | R&D Systems | clone 17611 | 1:100 |
| Antibody | TLR2 | BD Biosciences | clone 6C2 | 1:200 |
| Antibody | Mincle | Invivogen | clone 6G5 | 1:50 |
| Antibody | mαr IgG2b secondary | Invitrogen | ref SA5-10184 | 1:50 |
| Antibody | MHC-II/Iab | BD Bioscience | clone AF6-120.1 | 1:200 |
| Antibody | CD80 | eBioscience | clone 16–10 A1 | 1:100 |

*Continued on next page*

*Continued*

| Reagent type (species) or resource | Designation | Source or reference | Identifiers | Additional information |
|---|---|---|---|---|
| Antibody | CD86 | BD Bioscience | clone GL1 | 1:100 |
| Antibody | CD115 | eBioscience | clone AFS98 | 1:100 |
| Antibody | RANK/CD265 | eBioscience | clone R12-31 | 1:100 |
| Antibody | TNF | ThermoFisher Scientific | clone MP6-XT22 | 1:400 |
| Antibody | Cx3cr1 | ThermoFisher Scientific | clone 2A9-1 | 1:100 |
| Antibody | CD16/32 | eBioscience | clone 93 | 1:100 |
| Antibody | CD4 | BD Biosciences | clone RM4-5 | 1:1000 |
| Antibody | Phospho-Syk (Tyr348) | eBioscience | clone moch1ct | |
| Antibody | anti-Osterix | Abcam | ab22552 | dilution 1:800 |
| Antibody | anti-sclerostin | R&D System | AF1589 | dilution 1:200 |
| Sequence-based reagent | *36B4* | PCR primers | | TCCAGGCTTTGGGCATCA and CTTTATCAGCTGCACATCACTCAGA |
| Sequence-based reagent | *Syk* | PCR primers | | AACGTGCTTCTGGTCACACA and AGAACGCTTCCCACATCAGG |
| Sequence-based reagent | *Ctsk* | PCR primers | | CAGCAGAGGTGTGTACTATG and GCGTTGTTCTTATTCCGAGC |
| Sequence-based reagent | *Nfatc1* | PCR primers | | TGAGGCTGGTCTTCCGAGTT and CGCTGGGAACACTCGATAGG |
| Sequence-based reagent | si*Tlr2* | Horizon Discovery LTD | L-062838–02- 0005 | |
| Chemical compound, drug | Curdlan | Sigma-Aldrich | C7821 | |
| Chemical compound, drug | Zymosan | Sigma-Aldrich | Z4250 | |
| Chemical compound, drug | GlcC$_{14}$C$_{18}$ | InvivoGen | tlrl-gcc | |
| Chemical compound, drug | FITC dextran | Sigma-Aldrich | FD4 | |
| Chemical compound, drug | H33342 | Sigma-Aldrich | B2261 | 5 µg/mL |
| Commercial assay or kit | RatLaps (CTX-I) EIA | Immunodiagnostic Systems Limited | NC9787201 | |
| Commercial assay or kit | Annexin V Apoptosis detection kit | BD Biosciences | 556547 | |
| Commercial assay or kit | Acid Phosphatase Leucocyte kit | Sigma-Aldrich | 387 A | |
| peptide, recombinant protein | rmRANKL | R&D System | 462-TEC | 30 ng/mL |
| peptide, recombinant protein | rmM-CSF | R&D System | 416 ML | 25 ng/mL |
| peptide, recombinant protein | rmIL4 | Peprotech | 214–14 | 10 ng/mL |
| peptide, recombinant protein | rmGM-CSF | Peprotech | 315–03 | 10 ng/mL |
| Software, algorithm | ImageJ software version 1.53 | NIH, Bethesda, MD | | |
| Software, algorithm | FlowJo 10.8.1 | FlowJo | | |
| Software, algorithm | NRecon software | Bruker µCT, Belgium | | |
| Software, algorithm | Graph Pad Prism 9.4 software | Graph Pad Prism | | |

## Mice and ovariectomy-induced osteoporosis

C57BL/6 mice were purchased from Charles River Laboratory at 4 weeks of age and housed in the Animal Facility of the University Côte d'Azur. Animals were maintained under a 12-hr light/dark cycle, and food and tap water were provided ad libitum. Female 6-week-old C57BL/6 mice were randomly

divided into two groups for subsequent bilateral ovariectomy or SHAM surgery and were closely monitored until complete healing. Starting from 2 weeks after surgery, mice received by gavage *Sb* (Biocodex, Gentilly, France) 3 g/kg of body weight, 3 times per week until the end of the experiment. Six weeks after surgery, mice were sacrificed. *CD11cΔSyk mice and CD11c-Cre* littermates (*Iborra et al., 2012*) as well as *Clec4e*$^{+/+}$ and *Clec4e*$^{-/-}$ (B6.Cg-*Clec4e*$^{tm1.1Cfg}$) (*Wells et al., 2008*) female mice were bred in Centro Nacional de Investigaciones Cardiovasculares in SPF conditions. All experiments were approved by the French Ministry of Health, Higher Education and Research (autorization number 8389–2016121216457153 v3) and conducted in accordance with the Institutional Ethics Committee on Laboratory Animals (CIEPAL-Azur, Nice Sophia-Antipolis, France).

## Primary cell culture and osteoclast differentiation

i-OCLs and t-OCLs were differentiated in vitro as described previously (*Halper et al., 2021*; *Ibáñez et al., 2016*) from 6-week-old C57BL/6 mice (*Figure 1—figure supplement 1*). Briefly, CD11c$^+$ BM-derived DCs cells were obtained by culturing 5×10$^5$ BM cells/well in 24-well plates in RPMI medium (ThermoFisher Scientific) supplemented with 5% serum (Hyclone, GE Healthcare), 1% penicillin-streptomycin (ThermoFisher Scientific), 50 µM 2-mercaptoethanol (ThermoFisher Scientific), 10 ng/mL Gm-csf, and 10 ng/mL IL-4 (both PeproTech). The differentiation of t-OCLs and i-OCLs took 4–5 days and 5–6 days, respectively. CD11c$^+$ DCs were isolated using biotinylated anti-CD11c (1:200; clone HL3; BD Biosciences) and anti-biotin microbeads (Miltenyi Biotec). iOCLs were differentiated by seeding a total of 2×10$^4$ CD11c$^+$ DCs/well on 24-well plates in MEM-alpha (ThermoFisher Scientific) including 5% serum (Hyclone, GE Healthcare), 1% penicillin-streptomycin, 50 µM 2-mercaptoethanol, 25 ng/mL M-csf, and 30 ng/mL Rank-L (both R&D) (OCL differentiation medium). For t-OCL culture, 2×10$^5$ CD11b$^+$ monocytic BM cells that were isolated by biotinylated anti-CD11b (1:100; clone M1/70; ThermoFisher Scientific) and anti-biotin microbeads (Miltenyi Biotec) were seeded per well on 24-well plates in OCL differentiation medium as described above. For OCL differentiation from OVX and SHAM control mice, 5×10$^5$ BM cells were cultured per well in 24-well plates in OCL differentiation medium as indicated above. When indicated, differentiation was performed in the presence of indicated concentrations of curdlan, zymosan, or GlcC$_{14}$C$_{18}$. Medium conditioned by *Sb* CNCM I-745 (*Sb*-CM) was prepared by culture of the yeast in MEM-alpha overnight. The medium was collected and filtered through a sterile 22-µm filter, complemented with serum and antibiotics as described above and used as CM at indicated dilutions. All agonists and *Sb*-CM were added at day 0 (start of culture) and day 3 when the medium was changed. OCL differentiation was evaluated at the end of the differentiation after TRAcP staining according to manufacturer's instructions (Sigma-Aldrich). Mature OCLs were enumerated under a light microscope as multinucleated (≥three nuclei/cell) TRAcP$^+$ cells. OCL activity was evaluated by seeding OCL progenitors on plates coated with resorbable matrix (Osteo-assay) in OCL differentiation medium. All agonists and *Sb*-CM were added at day 3 when the medium was changed and OCLs started to fuse. Resorbed areas were quantified (ImageJ software version 1.53, NIH, Bethesda, MD) after removing of the cells with water and staining of the mineralized matrix with alizarin red.

## RNA-sequencing

RNAseq analysis was performed on mature multinucleated OCLs. After differentiation in vitro, t-OCLs and i-OCLs (five biological replicates, each derived from one mouse) were detached with Accutase, labeled with 5 µg/mL H33342, and sorted on their multinucleation as previously described (*Madel et al., 2018*; c.f. *Figure 1—figure supplement 1b* for the gating strategy). Total RNA (100 ng) was extracted from sorted OCLs (RNeasy kit, Qiagen), and directional libraries were prepared (Truseq stranded total RNA library kit, Illumina). Libraries were pooled and sequenced paired-ended for 2×75 cycles (Nextseq500 sequencer, Illumina). 30–40 million fragments were generated per sample, and quality controls were performed. Data were analyzed by two approaches as described previously (*Madel et al., 2020*). Both gave equivalent results. For the first one, reads were 'quasi' mapped on the reference mouse transcriptome (Gencode vM15) and quantified (SALMON software, mapping mode and standard settings) (*Patro et al., 2017*). Transcript count estimates and confidence intervals were computed using 1.000 bootstraps to assess technical variance. Transcript counts were aggregated for each gene for computing gene expression levels. Gene expression in biological replicates (n=5) was then compared between t-OCLs and i-OCLs using Phantassus and the limma R package (https://ctlab.

itmo.ru/phantasus) (*Daria Zenkova, 2018*). For the second approach, raw FASTQ reads were trimmed with Trimmomatic and aligned to the reference mouse transcriptome (Gencode mm10) with STAR (*Dobin et al., 2013*) on the National Institutes of Health high-performance computing Biowulf cluster. Gene-assignment and estimated read counts were assessed using HTseq (*Anders et al., 2015*). Gene expression was compared between t-OCLs and i-OCLs using DESeq2 (*Love et al., 2014*) with the Wald test (FDR <0.01).

## miRNA profiling and validation by RT-qPCR

Total RNA from sorted OCLs was extracted using the miRNeasy Micro Kit (QIAGEN), and the procedure automatized using the QIAcube (QIAGEN). The miRNA expression profiles were analyzed on paired samples using the TaqMan Array rodent MicroRNA Card Set v3.0 (TLDA, Applied Biosystems) after pre-amplification steps, according to manufacturer's instructions. Relative expression and statistical analysis were calculated using the ExpressionSuite software (Applied Biosciences), which included the student's t-test for sample group comparisons and built Volcano Plot comparing the size of the fold change (biological significance) to the statistical significance (p-value). Dysregulated miRNAs were examined with MirWalk (*Dweep and Gretz, 2015*), a miRNA database aiming to identify predicted and validated target genes and related pathways. This software provides information on miRNA-target interactions, not only on 3′-UTR, but also on the other regions of all known genes and simultaneously interrogates several algorithms (TargetScan, Miranda, RNA22, and miRWalk). We used a high predictive score with at least three of the four queried algorithms predicting miRNA target genes. Comparison of the expression patterns of 750 miRNAs in i-OCLs and t-OCLs was performed using Ct values <35, difference of at least twofold with a p-value <0.05.

Mature miRNAs of interest were specifically converted into cDNA using TaqMan microRNA reverse transcription kit according to the manufacturer's protocol (Applied Biosystems). RT-specific primers 5× (ThermoFisher) were multiplexed in a primer pool containing 1% of each diluted in an adequate volume of Tris-EDTA 1× . Pre-amplification step was performed using FAM-labeled specific PCR primers 20× and TaqMan PreAmp Master Mix kit (Applied Biosystems) for 12 cycles. Alternatively, these preliminary steps were performed using Megaplex RT and PreAmp Primers, Rodent pool A (Applied Biosystems) that include specific primers for miRNAs of interest. was performed on diluted pre-amp products using the specific TaqMan PCR primers and TaqMan Universal Master Mix II with no uracile N-glycosylase and run on Viia7 system (Applied Biosystems) in 96-well PCR plates for 40 cycles. Relative miRNA expression was normalized on sno202 expression in murine cells with the $2^{-\Delta CT}$ method.

## Bone structure analyses

Long bones of OVX and SHAM-operated mice were fixed in 4% paraformaldehyde. Bone microarchitecture analysis using high-resolution μCT was performed at the pre-clinical platform ECELLFRANCE (IRMB, Montpellier, France). Cortical and trabecular femora were imaged using high-resolution μCT with a fixed isotropic voxel size of 9 μm with X-ray energy of 50 kV, current of 500 mA, 0.5 mm aluminum filter, and 210 ms exposure time. Quantification of bone parameters was performed on the trabecular region of the proximal part of each femur (1.72 mm long) and on the cortical region (0.43 mm long region centered at the femoral midshaft) on CT Analyzer software (Bruker microCT, Belgium). For visual representation, 3D reconstructions were generated using NRecon software (Bruker μCT, Belgium).

## FITC-dextran permeability assay

At 1 hr before sacrifice, mice received oral gavage of 3–5 kDa FITC–dextran (Sigma-Aldrich) (60 mg/100 g body weight). FITC-dextran concentration in serum was measured by fluorometry in a fluorimeter (Xenius, SAFAS, Monaco) at 488/525 nm. Standard curve was prepared using dilutions of FITC-dextran in PBS with 20% fetal calf serum.

## TRAcP staining and immunohistochemistry

Tibias were fixed in 4% paraformaldehyde and decalcified for 72 hr in 4.13% EDTA, 0.2% paraformaldehyde pH 7.4, at 50°C in KOS microwave tissue processor (Milestone, Michigan, USA). They were then dehydrated and embedded in paraffin. TRAcP staining was performed as described (*Lézot et al.,*

*2015*) with Mayer hematoxylin counterstaining on 3-μm thick sections to identify OCLs and three images of four to five biological replicates were analyzed. Immunostaining of osteoblasts and osteocytes was performed with rabbit polyclonal anti-Osterix antibody (ab22552, dilution 1/800; Abcam, Cambridge, UK) and rabbit polyclonal anti-sclerostin antibody (AF1589, dilution1:200; R&D System, Abingdon, UK), respectively, with Gill2 hematoxylin counterstaining. For enumeration of osteoblasts and osteocytes, one image of four to five biological replicates was analyzed. Analyses were performed on the entire trabecular area axcluding the cortical area. Stained sections were automatically numerized (nanozoomer, Hamamatsu photonics) before observation (NDP view virtual microscope, Hamamatsu) and quantification (ImageJ software version 1.53, NIH, Bethesda, MD).

## Dosage of biochemical parameters

Concentration of lactate, propionate, and butyrate in the serum was determined by ion chromatography analysis after depletion of proteins and lipids with acetonitrile (Sigma-Aldrich). Samples were loaded on a Dionex ICS-5000 Plus system automatic device (ThermoScientific), and elution was performed according to the manufacturer's protocol. Chromatograms were aligned to standard solutions of each compound individually. Compound concentrations were determined using Chromeleon software (Thermo Scientific) by measuring surface area under the curve of the peaks and were compared to the corresponding ion standard profiles.

Serum crosslaps (CTX) were evaluated by enzyme-linked immunosorbent assay according to the manufacturer's protocol (RatLaps (CTX-I) EIA, Immunodiagnostic Systems Limited).

## Flow cytometry analysis

CD11b$^+$ monocytic BM cells and BM-derived CD11c$^+$ DCs cells were analyzed for their expression of Dectin-1 (1:100; clone bg1fpj; ThermoFisher Scientific), Dectin-2 (1:100, clone 17611; R&D Systems), Tlr2 (1:200; clone 6C2; BD Biosciences), and Mincle (1:50, clone 6G5, Invivogen).

For analysis of BM-derived DC maturation, BM-DCs were treated or not with curdlan or *Sb*-CM at the indicated dose for 24 hr in OCL differentiation medium (containing 25 ng/mL M-csf and 30 ng/mL Rank-L). They were then labeled with anti-CD11c (1:200; Clone N418, eBioscience), MHC-II/IAb (1:200; Clone AF6-120.1, BD Bioscience), CD80 (1:100; Clone 16–10 A1, eBioscience), CD86 (1:100; Clone GL1, BD Bioscience), CD115 (1:100; Clone AFS98, eBioscience), and Rank/CD265 (1:100; clone R12-31, eBioscience). Cells were analyzed by flow cytometry (BD FACSCanto-II, BD Bioscience).

To investigate the Syk phosphorylation after curdlan stimulation, BM-derived DCs were stimulated for 15 min in OCL differentiation medium with the indicated doses of curdlan and subsequently fixed with 2% PFA (Transcription Factor Fixation/Permeabilization kit, eBioscience) over night at 4°C. Surface staining was then performed with anti-MHC-II/IAb (1:100;Clone AF6-120.1, BD Bioscience) and CD11c (1:200; Clone N418, eBioscience) antibodies before the cells were permeabilized with 1× Saponine (1 g/100 mL) for 15 min and stained with Phospho-Syk (Tyr348, clone moch1ct, eBioscience) for 45 min. Cells were washed and acquired on a BD FACSCanto II.

For FACS analysis on OCLs, mature OCLs were detached using Accutase (ThermoFisher Scientific), labeled with 5 μg/mL H33342 and with anti-Dectin-1, Dectin-2, Tlr2, and Mincle antibodies and analyzed after doublet exclusion as multinucleated cells with three or more nuclei as previously described (*Madel et al., 2018*) for their expression of these markers (see *Figure 1—figure supplement 1b* for the gating strategy). Cells were analyzed by flow cytometry (BD FACSCanto-II, BD Bioscience).

For intracellular cytokine analysis, T cells isolated from the BM of SHAM and OVX mice were stimulated with phorbol myristate acetate (PMA), ionomycin, and brefeldin A, labeled with anti-CD4 antibody (1:1000; clone RM4-5; BD Biosciences) and fixed with 4% formaldehyde overnight as described (*Ciucci et al., 2015*). Cells were subsequently stained with anti-Tnf-α antibody (1:400; clone MP6-XT22; ThermoFisher Scientific) in 1× Saponine (1 g/100 mL). Data were acquired using a FACSCanto-II (BD Biosciences). All FACS data were analyzed with FlowJo 10.8.1.

## In vitro mineralization assay

Osteoblastic differentiation of BM-MSC cells from C57Bl/6 mice was performed for 3 weeks in MEM-alpha medium with 10% FBS, 1% penicillin-streptomycin, 50 μM b-mercaptoethanol supplemented with 170 μM L-ascorbic acid, 10 mM β-glycerophosphate, and 0.1 μM dexamethasone (Merck). This differentiation medium (with adjusted concentration of inductors) was supplemented with the

indicated % of *Sb*-CM. Medium was changed every 4 days. At day 21, cultures were fixed with 4% PFA and stained with 2% Alizarin red S (Merck). For the quantification, the Alizarin staining was dissolved in acetic acid (10%), heated 10 min at 85°C, centrifuged, and buffered with ammonium hydroxide. Absorbance at 405 nm was measured (Xenius SAFAS, Monaco).

### Blocking of Dectin-1, TLR2, and Mincle

For inhibition with siRNA, BM-derived DCs were transfected with ON-TARGET SMART pool of four siRNA to mouse *Tlr2* (50 nM, Dharmacon, Horizon Discovery, USA) using lipofectamine RNAiMAX (Invitrogen) and were further differentiated into OCLs for 5–6 days either in OCL differentiation medium in the presence of the indicated concentration of agonists, or in *Sb*-CM as described above.

For inhibition with blocking antibodies, BM-derived DCs were cultured either in OCL differentiation medium containing the indicated concentration of agonists, or in *Sb*-CM. Anti-Dectin-1 (clone bg1fpj; ThermoFisher Scientific), anti-Mincle (clone 6G5, Invivogen), or control isotype antibodies were added at the indicated concentration at the beginning of the differentiation. Medium was changed at day 3.

For both approaches, OCLs differentiation was evaluated after TRAcP staining, and mature OCLs were enumerated under a light microscope as multinucleated TRAcP[+] cells.

### RT-qPCR

BM-derived DCs were stimulated for 72 hr with the indicated doses of curdlan. Total RNA was extracted with Trizol according to the manufacturer protocol. RNAs were reverse transcribed (Superscript II, Life Technologies), and RT-PCR was performed using SYBR green and the primers indicated below. Results were normalized to the 36B4 gene with the $2^{-\Delta Ct}$ method. *Syk*: AACGTGCTTCTGGTCACACA and AGAACGCTTCCCACATCAGG; *Ctsk*: CAGCAGAGGTGTGTACTATG and GCGTTGTTCTTATTCCGAGC; *Nfatc1*: TGAGGCTGGTCTTCCGAGTT and CGCTGGGAACACTCGATAGG; *Rplp0*: TCCAGGCTTTGGGCATCA and CTTTATCAGCTGCACATCACTCAGA.

### Statistical analysis

Data were analyzed and statistics prepared using Graph Pad Prism 9.2 software. Analyses were done using two-tailed unpaired t-test when comparing two groups and ANOVA with multiple comparison test when comparison of more than two groups. Statistical significance was considered at $p < 0.05$, and experimental values are presented as mean ± SD. Biological replicates were obtained from different mice.

## Acknowledgements

The authors would like to acknowledge the Genomic Facility of the UFR Simone Veil, Université Versailles-Saint-Quentin (France) for the RNA sequencing, the IRCAN animal core facility that is supported by "la Région Provence Alpes-Côte d'Azur" (Nice, France) as well as the Montpellier preclinical platform of ECELLFRANCE for µCT analysis (IRMB, Montpellier, France). They also thank the Laboratory of Biochemistry-Hormonology (CHU, Nice, France) for CTX dosage and M Salah and D Carro (LP2M, Nice France) for their technical contribution. This work utilized the computational resources of the NIH-HPC-Biowulf cluster (http://hpc.nih.gov).

The work was supported by the Agence Nationale de la Recherche (ANR-16-CE14-0030) as well as by the French government, managed by the ANR as part of the Investissement d'Avenir UCA[JEDI] project (ANR-15-IDEX-01) and Biocodex (Gentilly, France). M-B M was supported by the Fondation pour la Recherche Médicale (FRM, ECO20160736019), and by awards from the Fondation Arthritis, the Société Française de Biologie des Tissus Minéralisés (SFBTM), the European Calcified Tissue Society (ECTS) and the American Society of Bone and Mineral Research (ASBMR). J.H. was supported by an award from ECTS, an award from SFBTM, and a fellowship from the Société Française de Rhumatologie (SFR). T.C. was supported by the Intramural Research Program of the National Cancer Institute, Center for Cancer Research, National Institutes of Health.

# Additional information

## Competing interests

Dorota Czerucka, Claudine Blin-Wakkach: received a research grant from Biocodex. Biocodex had no role in the design of the study, in the analysis and interpretation of the data and in the preparation of the manuscript. The other authors declare that no competing interests exist.

## Funding

| Funder | Grant reference number | Author |
|---|---|---|
| Agence Nationale de la Recherche | ANR-16-CE14-0030 | Henri-Jean Garchon<br>Henri-Jean Garchon |
| Fondation pour la Recherche Médicale | ECO20160736019 | Maria-Bernadette Madel |
| Agence Nationale de la Recherche | ANR-15-IDEX-01 | Maria-Bernadette Madel |

The funders had no role in study design, data collection and interpretation, or the decision to submit the work for publication.

## Author contributions

Maria-Bernadette Madel, Formal analysis, Investigation, Methodology, Validation, Writing – original draft, Writing – review and editing; Julia Halper, Formal analysis, Investigation, Methodology, Validation, Writing – review and editing; Lidia Ibáñez, Matthieu Rouleau, Formal analysis, Investigation, Methodology, Writing – review and editing; Lozano Claire, Formal analysis, Investigation, Supervision; Antoine Boutin, Formal analysis, Investigation, Methodology; Adrien Mahler, Formal analysis, Funding acquisition; Rodolphe Pontier-Bres, Majlinda Topi, Formal analysis; Thomas Ciucci, Conceptualization, Writing – review and editing, Writing – original draft; Christophe Hue, Writing – review and editing; Jerome Amiaud, Formal analysis, Investigation; Salvador Iborra, Writing – review and editing, Validation; David Sancho, Writing – review and editing, Investigation; Dominique Heymann, Conceptualization, Supervision, Writing – review and editing; Henri-Jean Garchon, Formal analysis, Funding acquisition, Supervision, Writing – review and editing, Provided and participated in experiments on CD11c-Syk mutant mice; Dorota Czerucka, Funding acquisition, Writing – review and editing, Provided and participated in experiments on CD11c-Syk mutant mice; Florence Apparailly, Formal analysis, Funding acquisition, Supervision, Writing – review and editing; Isabelle Duroux-Richard, Formal analysis, Investigation, Writing – review and editing; Abdelilah Wakkach, Conceptualization, Supervision, Methodology, Writing – review and editing; Claudine Blin-Wakkach, Conceptualization, Formal analysis, Funding acquisition, Methodology, Project administration, Supervision, Writing – original draft, Writing – review and editing

## Author ORCIDs

Julia Halper ![ORCID] http://orcid.org/0000-0003-3967-6633
Lidia Ibáñez ![ORCID] http://orcid.org/0000-0002-2217-8817
Matthieu Rouleau ![ORCID] http://orcid.org/0000-0002-0075-9880
Rodolphe Pontier-Bres ![ORCID] http://orcid.org/0000-0002-4139-5913
Thomas Ciucci ![ORCID] http://orcid.org/0000-0002-5828-0207
Claudine Blin-Wakkach ![ORCID] http://orcid.org/0000-0002-2621-3907

## Ethics

All experiments were approved by the french ministry of health, higher education and research (autorization number 8389-2016121216457153v3) and conducted in accordance with the Institutional Ethics Committee on Laboratory Animals (CIEPAL-Azur, Nice Sophia-Antipolis, France).

## Decision letter and Author response

Decision letter https://doi.org/10.7554/eLife.82037.sa1
Author response https://doi.org/10.7554/eLife.82037.sa2

## Additional files

### Supplementary files
• MDAR checklist

### Data availability
RNAseq data have been deposited in ENA (European Nucleotide Archive) accession number PRJEB42043.

The following dataset was generated:

| Author(s) | Year | Dataset title | Dataset URL | Database and Identifier |
|---|---|---|---|---|
| Madel MB, Garchon HJ, Wakkach A, Blin-Wakkach C | 2022 | Targeting inflammatory osteoclasts with probiotic yeast S. boulardii reduces bone loss in osteoporosis | https://www.ebi.ac.uk/ena/browser/view/PRJEB42043 | ENA, PRJEB42043 |

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
