## [Editor Report]

This important work substantially advances our understanding osteoclast diversity such as inflammatory and state osteoclasts in pathological condition, which was demonstrated by administration of the yeast probiotic Saccharomyces boulardi CNCM I-745 (Sb) in vivo reduced bone loss in OVX but not sham mice by reducing inflammatory osteoclasts, and Sb derivatives specifically inhibited directly the differentiation of inflammatory but not steady state osteoclasts in vitro. The evidence supports the conclusions, through combined transcriptomic profiling, differentiation assays and in vivo analysis in mice to decipher specific traits for inflammatory and steady state osteoclasts. The work will be of broad interest to Immunologists, Microbiologists and Cell biologists.

---

## [Decision Letter]

**Decision letter after peer review:**

Thank you for submitting your article "Specific targeting of inflammatory osteoclastogenesis by the probiotic yeast *S. boulardii* CNCM I-745 reduces bone loss in osteoporosis" for consideration by *eLife*. Your article has been reviewed by 3 peer reviewers, and the evaluation has been overseen by a Reviewing Editor and Mone Zaidi as the Senior Editor. The following individual involved in the review of your submission has agreed to reveal their identity: Marco Ponzetti (Reviewer #1).

The general objective of this work is the dissection of osteoclast diversity; in particular, the authors intend to identify the specific features and properties that distinguish inflammatory and steady-state (tolerogenic) osteoclasts. To this end, the authors perform a transcriptional analysis of inflammatory and tolerogenic osteoclasts and identify the pattern recognition receptors TLR2, Dectin-1, and Mincle as differentially expressed genes. Agonists of these receptors or yeast probiotics regulating the elicited mechanisms in vitro and in vivo caused a specific inhibition of the differentiation of inflammatory rather than tolerogenic osteoclasts, thus highlighting the preferential use of different differentiation pathways by the two distinct osteoclast populations.

The project is based on the previous knowledge and know-how of the authors on this peculiar skeletal cell population. The work is well conceived; the experiments are clearly designed and exploit state-of-the-art technologies. The results confirm the heterogeneity of osteoclasts and provide new insights in this respect. The in vitro and in vivo studies suggest that osteoclast heterogeneity can be purposedly modulated; which might be useful and advisable for therapeutic purposes. Overall, the work provides hints for further implementation and future broad applications to diseases featuring pathological bone loss. The osteoclasts are generated in vitro in the presence of M-CSF and RANKL to induce tolerogenic osteoclasts or GM-CSF / Il-4 to generate inflammatory osteoclasts. The demonstration of these cell populations in the S.b. treated mice in vivo is not present. The author tried to tackle this, by analyzing the differentiation potential of bone marrow progenitor cells of S.b. treated animals, which provides some information. The effect on tolerogenic osteoclasts could have been further evaluated, whether they are not affected at all, or whether there are also effects.

Essential revisions:

1. Please briefly describe biocodex in the disclosure section and provide details of the connection, including the fact that biocodex is one of the funding bodies of the work. 3g/kg thrice a week seems like quite a high dosage, do the authors expect Sb to be effective also at lower dosages in humans? The authors could probably provide references where high-dose probiotics in mice then proved effective at lower doses in humans in the Discussion section. Do the authors have any data on the effectiveness of Sb in IBD-dependent bone loss? Could any of the identified markers be used to sort mature inflammatory osteoclasts from tolerogenic osteoclasts? (see reviewer 1's comments)

2. It would be helpful to understand the localization of the different cell types in vivo. Could the authors use a co-staining approach for osteoclast markers and the PPRs to identify this subpopulation in vivo? That would certainly strengthen the message of the manuscript. (see reviewer 2's comments)

3. By evaluating the expression profiling it was not clear, whether an unfiltered GO analysis of DEGs would reveal C-type lectin receptor signaling as well, or whether other discriminating factors would be superior. As I understood in Figure 1e there was already a pre-selection of osteoclast/resorptive associated genes. (see reviewer 2's comments)

4. The functional studies of the inhibition of PPRs seemed to decrease also strongly in cell numbers in general (Figure 4 a-d), suggesting that also progenitor cells are affected. The authors should address this through the analysis of earlier time points. (see reviewer 2's comments)

5. There are already publications on PPRs and their function on osteoclasts, e.g. for the Mincle and Dectin-1 and in these publications, the effects are claimed to be more general on osteoclasts. In part, even opposite effects are reported. The effects of silencing of these receptors on the "tolerogenic cells" and the consequence on resorption should be therefore more clear. (see reviewer 2's comments)

6. The authors talk about decreased osteoclast numbers in the histomorphometry, but the osteoclast area is shown in Figure 2f. The authors should show the numbers of multinucleated cells as well. (see reviewer 2's comments)

7. The failure to rescue cortical parameters by S.b. is intriguing, the authors should discuss it. (see reviewer 2's comments)

8. There is an apparent strong trend that osteocytes are increased in supplementary figure 2c. The authors should discuss it since they also mention trends (p=0.09) elsewhere. (see reviewer 2's comments)

9. Page 9, lines 313-314: in line with the references here provided, the authors may quantify these cytokines in their experimental conditions. (see reviewer 3's comments)

10. Please show the level of gene silencing obtained with siRNA targeting TLR2 (and control siRNA as well). (see reviewer 3's comments)

11. In the supplementary figure 2, please provide also cortical porosity data (panel a) and representative images of Osterix and Sost IHC staining for the control groups SHAM and SHAM+Sb. Please detail how osteoblast and osteocyte count was performed (how many sections per mouse? Quantization on the entire tissue section or a number of fields?) (see reviewer 3's comments)

12. Pages 12-13, Primary cell culture and osteoclast differentiation: please specify the duration of the osteoclast differentiation assay and the frequency of PRR agonist supplementation into the culture. Also, please indicate at which time point OCL activity was measured. (see reviewer 3's comments)

13. In general, we suggest the authors present all the data as scatter plots with bars (where applicable), as already done in a number of panels. (see reviewer 3's comments)

14. A point-by-point response to all three reviewers' comments.

*Reviewer #1 (Recommendations for the authors):*

The report presented by Madel/Halper and colleagues is very well-written, logical, easy to read, and with a very good flow and pace. All the experiments presented are scientifically sound and the conclusions are not overstated. As a person working in the field, I agree with the authors that osteoclast heterogeneity is an under-investigated and underestimated issue and that more research is needed to understand how the different sub-populations of osteoclasts participate in bone loss in pathological conditions. In this report, the authors first explore osteoclast heterogeneity using RNAseq and miRNA analyses, and then, find that specific pattern-recognition receptors are overexpressed in inflammatory osteoclasts compared to tolerogenic osteoclasts, and that this difference can also be observed in vivo in OVX mice, they proceed to administer a commercially-available strain of the probiotic yeast S. boulardii to OVX mice, observing prevention of the ovariectomy-induced bone loss, with an effect that is not oestrogen-mediated since uterus weight is comparable between treated and untreated mice. The report deserves publication in *eLife*, I only have a few comments that I hope will improve the quality of the manuscript even more.

Figure 2d type size is inconsistent and in general in the same figure sometimes the font size varies.

Figure 4e 10μg goes inside the space of the figure.

Supp figure 2b missing measuring units for obs and ocs.

Line 400 should read {greater than or equal to}.

Line 468 please specify how the region of interest was selected and the software and algorithms used for analysis (e.g. bruker CTan, thresholding method global).

Line 479 for how long?

A table collecting all antibodies used (also with RRID) and relative concentrations would be helpful as a supplementary. The same could be said for primer sequences/Taqman assay ID.s.

Line 533 what is the concentration of saponin 1X?

Please briefly describe biocodex in the disclosure section and provide details of the connection, including the fact that biocodex is one of the funding bodies of the work.

3g/kg thrice a week seems like quite a high dosage, do the authors expect Sb to be effective also at lower dosages in humans? The authors could probably provide references where high-dose probiotics in mice then proved effective at lower doses in humans in the Discussion section.

Do the authors have any data on the effectiveness of Sb in IBD-dependent bone loss?

Could any of the identified markers be used to sort mature inflammatory osteoclasts from tolerogenic osteoclasts?

Line 241 maybe the authors could show this data in a supplementary figure.

Did the authors evaluate the serum levels of IL4 and IL10 in OVX mice treated with Sb? Up to the authors' discretion but it would be interesting to know this, since if they are affected, they may play a part in the phenotype rescue observed, and if they are not, it would be even more interesting because everything would be caused by β-glucans or other compounds contained in the probiotics.

Were there any side effects to the probiotic treatment?

Figure 2D and all others: authors should also show the statistical comparisons between Sham-Sb and OVX-Sb.

Usually, osteoclasts are evaluated histomorphometrically using Oc.S/BS and Oc.N/BS – is there a specific reason why the authors used Tracp+ Area/Bone surface for these analyses?

It would be interesting to perform GSEA on these datasets as well, which might lead to further findings and useful data for the authors.

Again, congratulations to the authors for an interesting and well-done report.

*Reviewer #2 (Recommendations for the authors):*

1. It would be helpful to understand the localization of the different cell types in vivo. Could the authors use a co-staining approach for osteoclast markers and the PPRs to identify this subpopulation in vivo? That would certainly strengthen the message of the manuscript.

2. By evaluating the expression profiling it was not clear, whether an unfiltered GO analysis of DEGs would reveal C-type lectin receptor signaling as well, or whether other discriminating factors would be superior. As I understood in Figure 1e there was already a pre-selection of osteoclast/resorptive associated genes.

3. The functional studies of the inhibition of PPRs seemed to decrease also strongly in cell numbers in general (Figure 4 a-d), suggesting that also progenitor cells are affected. The authors should address this through the analysis of earlier time points.

4. There are already publications on PPRs and their function on osteoclasts, e.g. for the Mincle and Dectin-1 and in these publications, the effects are claimed to be more general on osteoclasts. In part, even opposite effects are reported. The effects of silencing these receptors on the "tolerogenic cells" and the consequence on resorption should be therefore more clear.

5. The authors talk about decreased osteoclast numbers in the histomorphometry, but the osteoclast area is shown in Figure 2f. The authors should show the numbers of multinucleated cells as well.

6. The failure to rescue cortical parameters by S.b. is intriguing, the authors should discuss it.

7. There is an apparent strong trend that osteocytes are increased in supplementary figure 2c. The authors should discuss it since they also mention trends (p=0.09) elsewhere.

8. In Figure 3a-d single data points are missing and should be added.

*Reviewer #3 (Recommendations for the authors):*

Suggestions to the authors:

– Additional data:

Page 9, lines 313-314: in line with the references here provided, the authors may quantify these cytokines in their experimental conditions.

Please show the level of gene silencing obtained with siRNA targeting TLR2 (and control siRNA as well).

In supplementary figure 2, please provide also cortical porosity data (panel a) and representative images of Osterix and Sost IHC staining for the control groups SHAM and SHAM+Sb. Please detail how osteoblast and osteocyte count was performed (how many sections per mouse? Quantization on the entire tissue section or a number of fields?).

– Method description:

Pages 12-13, Primary cell culture and osteoclast differentiation: please specify the duration of the osteoclast differentiation assay and the frequency of PRR agonist supplementation into the culture. Also, please indicate at which time point OCL activity was measured.

– Figure legends: Actually, we think referring to the supplementary figures in the legend to the main figures is confounding. We think the legend should describe only what is displayed in the corresponding figure.

– Text edits:

Page 3, line 60: HETEROGENEOUS instead of ETEROGENEOUS; line 78: i-OCLs (plural) instead of i-OCL; line 87: you can use the acronym i-OCLs here since it has been already introduced earlier.

Page 4, line 108: maybe use ANALYSIS instead of APPROACH; line 110: "and representing": we think AND can be deleted; line 112: "between BOTH OCL subsets": we think "between THE TWO OCL subsets" is more appropriate.

Page 5, line 150: IN HERE should be HEREIN; line 155: SHAM OCLs: we would prefer "OCLs derived from SHAM mice".

Page 7, line 219: "to a lesser extent" this can be deleted from our point of view; in fact, in both treatment groups (SHAM and OVX) the OCL % seems to be reduced by 50%; line 221: the term KINASE can be deleted here.

Page 8, line 247: the proportion OF; line 249: "Lastly,…(Figure 4g)" we are not convinced this statement is correctly placed here, since it refers to the result of an experiment described earlier, while the present paragraph deals with the effect of Sb-CM.

Page 9, line 283: "recognized by" can be replaced by FOR, in line with the construction used in the previous line; line 293: "in A several OF gastrointestinal disorders" should be "in several gastrointestinal disorders", or "in a SERIES (VARIETY or similar term) of"; line 312: "Sb IT has been described" delete IT.

Page 10, line 344: use DEMONSTRATED instead of IDENTIFIED; line 349: please check this line.

Page 12, line 365: "assigned into two groups" we suggest "DIVIDED INTO two groups"; lines 371-372: "french ministry of health. Higher education and research" all initials in capital letters.

Page 13, line 404 ALIZARINE should be ALIZARIN.

Page 15, line 499: SERIC should be SERUM.

Page 24, line 834: "FACS plots" should be "histograms".

Page 25, line 855: if we get it correct from the figure, it seems that the number of replicates is n{greater than or equal to}5 (not n=5); lines 860-861: n=?. Figure 4, panels a-c: please indicate in the legend what is represented in the graphs: mean {plus minus} s.d.? In the text and in the figures please confirm to the unit mL (not ml; at present, both are used here).

Figure 4, panel b: ZYMOZAN should be ZYMOSAN; panel h: it is not necessary to indicate the name of the specific PRR agonist used on top of each graph because it is indicated under each graph.

Figure 5, panel c and Figure 6, panel d: MHC2 should be MHC-II.

Legend to supplementary figure 2: invert cortical bone surface/bone volume and cortical thickness, in accordance with the order of the graphs in the figure; the last line: conditionned should be conditioned.

– Figures:

In general, we suggest the authors present all the data as scatter plots with bars (where applicable), as already done in a number of panels.

Figure 1, panel j, first graph: normalized to…?

Figure 2: we think it could be interesting to see a representative plot, to visualize double positive cells.

Figure 3, panels e and f: use "SHAM+Sb" and "OVX+Sb", in line with the other panels and figures.

---

## [Author Response]

Reviewer #1 (Recommendations for the authors):The report presented by Madel/Halper and colleagues is very well-written, logical, easy to read, and with a very good flow and pace. All the experiments presented are scientifically sound and the conclusions are not overstated. As a person working in the field, I agree with the authors that osteoclast heterogeneity is an under-investigated and underestimated issue and that more research is needed to understand how the different sub-populations of osteoclasts participate in bone loss in pathological conditions. In this report, the authors first explore osteoclast heterogeneity using RNAseq and miRNA analyses, and then, find that specific pattern-recognition receptors are overexpressed in inflammatory osteoclasts compared to tolerogenic osteoclasts, and that this difference can also be observed in vivo in OVX mice, they proceed to administer a commercially-available strain of the probiotic yeast S. boulardii to OVX mice, observing prevention of the ovariectomy-induced bone loss, with an effect that is not oestrogen-mediated since uterus weight is comparable between treated and untreated mice. The report deserves publication in eLife, I only have a few comments that I hope will improve the quality of the manuscript even more.

We thank the reviewer for it's positive comments.

Figure 2d type size is inconsistent and in general in the same figure sometimes the font size varies.Figure 4e 10μg goes inside the space of the figure.Supp figure 2b missing measuring units for obs and ocs.

All these points have been corrected in the corresponding figures.

Line 400 should read {greater than or equal to}.

This has been corrected.

Line 468 please specify how the region of interest was selected and the software and algorithms used for analysis (e.g. bruker CTan, thresholding method global).

The requested information has been added in the material and methods section*.*

Line 479 for how long?

The information has been added in the material and methods section.

A table collecting all antibodies used (also with RRID) and relative concentrations would be helpful as a supplementary. The same could be said for primer sequences/Taqman assay ID.s.

The key resources table has been provided with the revised manuscript.

Line 533 what is the concentration of saponin 1X?

The concentration is 1 g/100 mL. The information has been added in the material and method.

Please briefly describe biocodex in the disclosure section and provide details of the connection, including the fact that biocodex is one of the funding bodies of the work.

As requested, in the competing interest section we have indicated that Biocodex is a pharmaceutical laboratory and we have provided the requested information.

3g/kg thrice a week seems like quite a high dosage, do the authors expect Sb to be effective also at lower dosages in humans? The authors could probably provide references where high-dose probiotics in mice then proved effective at lower doses in humans in the Discussion section.

We agree with the reviewer that the dosage of Sb we used in vivo (~65 mg/mouse corresponding to ~3.10^9^ CFU/mouse/day, 3 times a week for 4 weeks) is much higher than the one used in human (200 mg/person/day, corresponding to ~10^10^ CFU). However, this range of dose is frequently used in studies evaluating the effects of Sb on chronic diseases in mouse (as for instance in 2 diabetic models in which mice received 0.5×10^8^ CFU of Sb/mouse/day for 8 weeks) (https://doi.org/10.1016/j.lfs.2022.120616) or 120mg/mouse/day for 4 weeks (DOI: 10.1128/mBio.01011-14).

Interestingly, in the context of osteoporosis, equivalent differences in the dosage of bacterial probiotics used in human and mouse have been reported. For instance, *Lactobacillus* strains were shown to reduce bone loss in osteoporotic at 1x10^9^ CFU/ml of drinking water/day (corresponding to about 4,5x10^9^ CFU/mouse/day) for 6 weeks, and this protective effect has been confirmed in human with a dose of 10^10^ CFU/ patient /day for 1 year (https://doi.org/10.1016/S2665-9913(19)30068-2). Same result was reported for *Lactobacillus* reuteri which is protective in mouse at a dose of 3x10^8^ CFU/3 times a week for 4 weeks (https://doi.org/10.1002/jcp.24636) and which reduces bone loss in osteoporotic patients at dose of 1x10^10^ CFU/patient/day for 1 year (DOI: 10.1111/joim.12805 ).

Therefore, based on such studies showing protective effect of probiotics with much lower doses in human compared to mouse, we can assume that lower doses of Sb would also be effective in human, even though the clear answer to this question would require a clinical trial in osteoporotic patients. This has been discussed in the Discussion section of our revised manuscript.

Do the authors have any data on the effectiveness of Sb in IBD-dependent bone loss?

Unfortunately, we don't have any data on the effect of Sb on IBD-dependent bone loss.

Could any of the identified markers be used to sort mature inflammatory osteoclasts from tolerogenic osteoclasts?

From our transcriptomic data we indeed identified markers, including those described here, that we are using to sort and analyse inflammatory and non-inflammatory osteoclasts for new specific properties. However, these new data will be the subject of another manuscript that is in preparation. Therefore we prefer not to include them in our manuscript as we believe it's not essential for this study. We hope the reviewer will understand.

Line 241 maybe the authors could show this data in a supplementary figure.

The requested data on cell apoptosis have been included in Figure 4—figure supplement 1a-c.

Did the authors evaluate the serum levels of IL4 and IL10 in OVX mice treated with Sb? Up to the authors' discretion but it would be interesting to know this, since if they are affected, they may play a part in the phenotype rescue observed, and if they are not, it would be even more interesting because everything would be caused by β-glucans or other compounds contained in the probiotics.

We performed the requested experiment by dosing serum concentration of IL-10 and IL-4 by ELISA. The results showed no significant difference in IL-10 concentration between OVX and OVX+Sb mice (see Author response image 1). This result has been added as data not shown. IL-4 was not detected in any of the conditions.

**Author response image 1. sa2fig1:** 

Were there any side effects to the probiotic treatment?

We did not observed any side effects of the probiotic in the treated mice.

Figure 2D and all others: authors should also show the statistical comparisons between Sham-Sb and OVX-Sb.

Figures 2 and 3 have been modified as requested to show the statistical comparisons between Sham-Sb and OVX-Sb.

Usually, osteoclasts are evaluated histomorphometrically using Oc.S/BS and Oc.N/BS – is there a specific reason why the authors used Tracp+ Area/Bone surface for these analyses?

In our hands, TRAcP staining in vivo is usually very intense, which makes it impossible to always see the limit between 2 OCLs and all nuclei, and thus to properly count individually these cells. That's why in all our previous studies, we have always evaluated OCL area and not their number. This approach has already been validated in a number of publications (for instance *=* Mansour et al., Cell research 2011. DOI:10.1038/cr.2011.21; Mansour et al., J Exp Med 2012. DOI:10.1084/jem.20110994; Moukengue B et al. EBioMedicine. 2020. doi: 10.1016/j.ebiom.2020.102704; Guihard P et al. Am J Pathol. 2015. doi: 10.1016/j.ajpath.2014.11.008; Lamoureux F, et al. Nat Commun. 2014. doi: 10.1038/ncomms4511; Gobin B, et al. Int J Cancer. 2015. doi: 10.1002/ijc.29040; Gobin B et al. Cancer Lett. 2014. doi: 10.1016/j.canlet.2013.11.017; Velasco CR, et al. Glycobiology. 2011. doi: 10.1093/glycob/cwr002).

However, to comply with the reviewer's comment, we modified the text not to talk about OCL number.

It would be interesting to perform GSEA on these datasets as well, which might lead to further findings and useful data for the authors.

This analysis has been performed (see Author response table 1). As it gave results equivalent to those presented in figure 1c, we did not include it in the revised manuscript.

**Author response table 1. sa2table1:** 

Gene Set Name	# Genes in Gene Set (K)	# Genes in Overlap (k)	k/K	p-value	FDR q-value
DEFENSE_RESPONSE	1816	109	0.0600	3.16E-47	3.88E-43
REGULATION_OF_IMMUNE_SYSTEM_PROCESS	1642	101	0.0615	1.54E-44	9.45E-41
INFLAMMATORY_RESPONSE	770	69	0.0896	2.09E-40	8.58E-37
CELL_ADHESION	1480	84	0.0568	2.77E-34	8.53E-31
IMMUNE_RESPONSE	1920	94	0.0490	2.27E-33	5.57E-30
CELL_ACTIVATION	1273	77	0.0605	3.16E-33	6.48E-30
BIOLOGICAL_PROCESS_INVOLVED_IN_INTERSPECIES_INTERACTION_BETWEEN_ORGANISMS	1835	91	0.0496	1.07E-32	1.89E-29
REGULATION_OF_CELL_ADHESION	811	60	0.0740	1.46E-30	2.24E-27
POSITIVE_REGULATION_OF_MULTICELLULAR_ORGANISMAL_PROCESS	1722	85	0.0494	2.39E-30	3.26E-27
REGULATION_OF_RESPONSE_TO_EXTERNAL_STIMULUS	992	65	0.0655	5.04E-30	6.19E-27

Again, congratulations to the authors for an interesting and well-done report.

We thank the reviewer for it's positive comments.

Reviewer #2 (Recommendations for the authors):1. It would be helpful to understand the localization of the different cell types in vivo. Could the authors use a co-staining approach for osteoclast markers and the PPRs to identify this subpopulation in vivo? That would certainly strengthen the message of the manuscript.

We agree with the reviewer that this is an interesting question. However, we have a new article in preparation in which the specific localization and new properties of inflammatory and tolerogenic osteoclasts will be analyzed in detail. Thus, we prefer not to include part of these new data in the present manuscript so as not to limit the interest of this new paper. We hope the reviewer will understand our position.

2. By evaluating the expression profiling it was not clear, whether an unfiltered GO analysis of DEGs would reveal C-type lectin receptor signaling as well, or whether other discriminating factors would be superior. As I understood in Figure 1e there was already a pre-selection of osteoclast/resorptive associated genes.

The reviewer is right, the analysis in former Figure 1e is done on a list of genes preselected from the Kegg pathway mmu04380 (osteoclast differentiation). In line with the reviewer's suggestion, we performed similar analysis on the totality of the genes differentially express between MN-OCLs and DC-OCLs, which also highlighted the c-lectin like receptor pathway. This new analysis has been added in Figure 1d.

3. The functional studies of the inhibition of PPRs seemed to decrease also strongly in cell numbers in general (Figure 4 a-d), suggesting that also progenitor cells are affected. The authors should address this through the analysis of earlier time points.

As requested by the reviewer, we performed a viability assay by measuring the apoptotic cells (Annexin –V+ PI+) by FACS 24 and 48 hours after the addition of agonists. Our results revealed that neither the agonists, nor the Sb-conditioned medium, induced cell apoptosis. These results have been added in Figure 4—figure supplement 1a-d.

4. There are already publications on PPRs and their function on osteoclasts, e.g. for the Mincle and Dectin-1 and in these publications, the effects are claimed to be more general on osteoclasts. In part, even opposite effects are reported. The effects of silencing these receptors on the "tolerogenic cells" and the consequence on resorption should be therefore more clear.

We fully agree with the comment of reviewer. It's true that in the literature, it has been reported that curdlan reduces global osteoclast differentiation through Dectin-1. However, this effect is observed only with high concentrations (range of 10 µg/mL and more) of curdlan (https://doi.org/10.18632/oncotarget.18411) whereas the effect we observed specifically on i-OCLs is already achieved with 10 ng/mL. It has also been reported that the Dectin-1 protein is expressed in BM OCL progenitors but at low levels (https://doi.org/10.1074/jbc.M114.551416) and that Dectin-1 activation by curdlan efficiently reduces OCL formation from RAW cells only when cells are transfected with Dectin-1 (https://doi.org/10.1074/jbc.M114.551416). Although these studies have already been discussed in our manuscript, we have expanded this point in the revised version (see "discussion").

Regarding Mincle, we also agree that opposite effects have been reported in the literature and indeed stimulation of Mincle by necrotic osteocytes stimulates OCL differentiation (*doi: 10.1172/JCI134214*). We therefore performed in vitro experiments on BM cells from Mincle KO mice and microscanner analysis of these mice. For the microscanner, we obtained results similar to those of Andreev et al. 2020, with an increased bone mass in Mincle (Clec4e) KO mice. However, for the in vitro experiments, our results are divergent with this study. We performed osteoclastogenic differentiation in vitro from total BM cells, BM-derived DCs and BM monocytes, and we observed a tendency to an increased osteoclastogenesis, whatever the OCL progenitor we used. This discrepancy may be explain by the use of a different protocol for osteoclastogenesis as we start the differentiation adding M-CSF and RANKL together, while Andreev et al. 2020 add M-CSF first and 2 days later they add RANKL. Furthermore, we tested the effect of the Mincle agonist GlcC14C18 on osteoclastogenesis from Mincle KO DCs and our results show that the inhibition of osteoclastogenesis is reduced compared to WT confirming the implication of Mincle in this inhibition.

Therefore, these observations revealed the complex effects of Mincle on osteoclastogenesis, as already shown on other monocytic cells. These new results have been added in in Figure 4-Suppl Figure 2 and commented in the discussion.

Lastly, as requested, we performed additional experiment to assess the effect of agonist on the resorption capacity of tolerogenic OCLs, which did not reveal any significant effect. The data were added in Figure 4—figure supplement 1.

5. The authors talk about decreased osteoclast numbers in the histomorphometry, but the osteoclast area is shown in Figure 2f. The authors should show the numbers of multinucleated cells as well.

As explained for reviewer 1, in our hands, TRAcP staining in vivo is usually very intense, which makes it impossible to always see the limit between 2 OCLs and all nuclei, and thus to properly count individually these cells. That's why in all our previous studies, we have always evaluated OCL area and not their number. This approach has already been validated in a number of publications (for instance = Mansour et al., Cell research 2011. DOI:10.1038/cr.2011.21; Mansour et al., J Exp Med 2012. DOI:10.1084/jem.20110994; Moukengue B et al. EBioMedicine. 2020. doi: 10.1016/j.ebiom.2020.102704; Guihard P et al. Am J Pathol. 2015. doi: 10.1016/j.ajpath.2014.11.008; Lamoureux F, et al. Nat Commun. 2014. doi: 10.1038/ncomms4511; Gobin B, et al. Int J Cancer. 2015. doi: 10.1002/ijc.29040; Gobin B et al. Cancer Lett. 2014. doi: 10.1016/j.canlet.2013.11.017; Velasco CR, et al. Glycobiology. 2011. doi: 10.1093/glycob/cwr002).

However, to comply with the reviewer's comment, we modified the text not to talk about OCL number.

6. The failure to rescue cortical parameters by S.b. is intriguing, the authors should discuss it.

Such a lack of effect on cortical bone has also been reported in OVX mice treated with bacterial probiotics (*McCabe et al., 2013, DOI 10.1002/jcp.24340; Britton et al., 2014, DOI 10.1002/jcp.24636 ; Lawenius et al., 2022, DOI 10.1530/JOE-21-0408* …). The reasons are not clear but it could be due to a too short period of treatment, a starting point too late after surgery, or a preferential location of certain OCL subtypes in the trabecular versus cortical bone.

This point has been commented in the "results" section.

7. There is an apparent strong trend that osteocytes are increased in supplementary figure 2c. The authors should discuss it since they also mention trends (p=0.09) elsewhere.

The comparison of osteocyte numbers between the OVX and OVX+Sb groups yielded a p=0.22, far from significance. In contrast, the pVal mentioned by the reviewer (Figure 3c-d) are lower, despite not reaching significance (p=0.09 and 0.08 for butyrate and lactate, respectively). Moreover, the different metabolites we measured all showed a similar pattern of variation, in particular an increase after Sb treatment in OVX mice. It's because they all varied the same way that we discussed this tendency in this figure.

8. In Figure 3a-d single data points are missing and should be added.

Single data points have been added in all the figures.

Reviewer #3 (Recommendations for the authors):Suggestions to the authors:– Additional data:Page 9, lines 313-314: in line with the references here provided, the authors may quantify these cytokines in their experimental conditions.

As explained for reviewer 1, We performed the requested experiment by dosing by ELISA serum concentration of IL-10 and IL-4. The results showed no significant difference in IL-10 concentration between OVX and OVX+Sb mice (see Author response image 1). This result has been added as data not shown.

IL-4 was not detected in any of the conditions.

Please show the level of gene silencing obtained with siRNA targeting TLR2 (and control siRNA as well).

As requested, these data have been added in Figure 4—figure supplement 1e.

In supplementary figure 2, please provide also cortical porosity data (panel a) and representative images of Osterix and Sost IHC staining for the control groups SHAM and SHAM+Sb. Please detail how osteoblast and osteocyte count was performed (how many sections per mouse? Quantization on the entire tissue section or a number of fields?).

Unfortunately, it is not possible to provide data on cortical porosity with the parameters that have been recorded in the µCT facility. And we don't have anymore bone sample left to repeat the anaysis for cortical porosity. We hope the reviewer will understand.

The images of Osterix and Sost staining has been added for SHAM and SHAM+Sb groups in Figure 2-supplement figure 2. The requested information on cell count and quantization has been added in the material and method section.

– Method description:Pages 12-13, Primary cell culture and osteoclast differentiation: please specify the duration of the osteoclast differentiation assay and the frequency of PRR agonist supplementation into the culture. Also, please indicate at which time point OCL activity was measured.

The requested information has been added in the material and method section.

– Figure legends: Actually, we think referring to the supplementary figures in the legend to the main figures is confounding. We think the legend should describe only what is displayed in the corresponding figure.

This is a request of the journal. It is stated in the author guide that "Figure Supplements should be referred to in the legend of the associated primary figure".

Page 7, line 219: "to a lesser extent" this can be deleted from our point of view; in fact, in both treatment groups (SHAM and OVX) the OCL % seems to be reduced by 50%; line 221: the term KINASE can be deleted here.

We removed "to a lesser extend". The other points have been modified in the text.

Page 8, line 247: the proportion OF; line 249: "Lastly,…(Figure 4g)" we are not convinced this statement is correctly placed here, since it refers to the result of an experiment described earlier, while the present paragraph deals with the effect of Sb-CM.

This sentence also refers to Sb-CM, which, we agree, was not clearly indicated. This has been corrected. Morevoer, we deleted the SB-CM data from figure 4g to put it as a new figure (6b) to have all the data on Sb-CM in the same figure.

Page 25, line 855: if we get it correct from the figure, it seems that the number of replicates is n{greater than or equal to}5 (not n=5); lines 860-861: n=?. Figure 4, panels a-c: please indicate in the legend what is represented in the graphs: mean {plus minus} s.d.? In the text and in the figures please confirm to the unit mL (not ml; at present, both are used here).Figure 4, panel b: ZYMOZAN should be ZYMOSAN; panel h: it is not necessary to indicate the name of the specific PRR agonist used on top of each graph because it is indicated under each graph.Figure 5, panel c and Figure 6, panel d: MHC2 should be MHC-II.Legend to supplementary figure 2: invert cortical bone surface/bone volume and cortical thickness, in accordance with the order of the graphs in the figure; the last line: CONDITIONNED should be CONDITIONED.

All these points have been modified in the text and figures.

– Figures:In general, we suggest the authors present all the data as scatter plots with bars (where applicable), as already done in a number of panels.

We followed the reviewer's recommendation and presented the data as requested.

Figure 1, panel j, first graph: normalized to…?

"*Normalized to mode*" is a classical representation of FACS data on histograms. It's a normalization made by the sofware (FlowJo) to harmonize the height of the histograms that could change among the graphs depending on the number of cells that have been acquired in each sample.

Figure 2: we think it could be interesting to see a representative plot, to visualize double positive cells.

Unfortunately, in these experiments the markers were analyzed independently so it's not possible to show double positive cells.